# Metastasis of colon cancer requires Dickkopf-2 to generate cancer cells with Paneth cell properties

Jae Hun Shin[1,2]*[†], Jooyoung Park[3][†], Jaechul Lim[4], Jaekwang Jeong[5], Ravi K Dinesh[6], Stephen E Maher[7], Jeonghyun Kim[2], Soyeon Park[2], Jun Young Hong[8], John Wysolmerski[5], Jungmin Choi[3], Alfred LM Bothwell[9,10]*

[1]Integrative Science and Engineering Division, Underwood International College, Yonsei University, Incheon, Republic of Korea; [2]Institute of Advanced Bio-Industry Convergence, Yonsei University, Seoul, Republic of Korea; [3]Department of Biomedical Sciences, Korea University College of Medicine, Seoul, Republic of Korea; [4]College of Veterinary Medicine, Seoul National University, Seoul, Republic of Korea; [5]Internal Medicine, Yale University School of Medicine, New Haven, United States; [6]Department of Pathology, Stanford University, Stanford, United States; [7]Department of Urology, Yale University School of Medicine, New Haven, United States; [8]Department of Systems Biology, College of Life Science and Biotechnology, Yonsei University, Seoul, Republic of Korea; [9]Department of Pathology, Microbiology and Immunology, University of Nebraska Medical Center, Omaha, United States; [10]Department of Immunobiology, Yale University School of Medicine, New Haven, United States

*For correspondence:
jaehun.shin@yonsei.ac.kr (JHunS);
albothwell@unmc.edu (ALMB)

[†]These authors contributed equally to this work

## eLife Assessment

This **valuable** study proposes that protein secreted by colon cancer cells induces cells with Paneth-like properties that favor colon cancer metastasis. The evidence supporting the conclusions is **solid** but the study would benefit from more direct experiments to test the functional role of Paneth-like cells and to monitor metastasis from colon tumors. The work will be of interest to researchers studying colon cancer metastasis.

**Abstract** Metastasis is the leading cause of cancer-related mortality. Paneth cells provide stem cell niche factors in homeostatic conditions, but the underlying mechanisms of cancer stem cell niche development are unclear. Here, we report that Dickkopf-2 (DKK2) is essential for the generation of cancer cells with Paneth cell properties during colon cancer metastasis. Splenic injection of *Dkk2* knockout (KO) cancer organoids into C57BL/6 mice resulted in a significant reduction of liver metastases. Transcriptome analysis showed reduction of Paneth cell markers such as lysozymes in KO organoids. Single-cell RNA sequencing analyses of murine metastasized colon cancer cells and patient samples identified the presence of lysozyme positive cells with Paneth cell properties including enhanced glycolysis. Further analyses of transcriptome and chromatin accessibility suggested hepatocyte nuclear factor 4 alpha (HNF4A) as a downstream target of DKK2. Chromatin immunoprecipitation followed by sequencing analysis revealed that HNF4A binds to the promoter region of *Sox9*, a well-known transcription factor for Paneth cell differentiation. In the liver metastatic foci, DKK2 knockout rescued HNF4A protein levels followed by reduction of lysozyme positive cancer cells. Taken together, DKK2-mediated reduction of HNF4A protein promotes the generation of lysozyme positive cancer cells with Paneth cell properties in the metastasized colon cancers.

## Introduction

Metastasis is the main cause of cancer-related death (*Dillekås et al., 2019*). Mutations in the *Apc* gene in intestinal epithelial cells cause adenoma formation by dys-regulation of Wnt signaling (*Rowan et al., 2000*). Accumulating oncogenic mutations in *Kras, Trp53,* and *Smad4* genes facilitate colorectal carcinogenesis and metastasis (*van Houdt et al., 2010*; *Drost et al., 2015*; *Fumagalli et al., 2017*). Recent studies have identified that *Lgr5* positive cells are required for metastasis in the murine models of colorectal cancer (*de Sousa e Melo et al., 2017*; *Fumagalli et al., 2020*). *Lgr5* positive cells include stem cells, transit amplifying cells and Paneth cells (*Sato et al., 2009*). Ablation of *Lgr5* positive cells in the primary tumor in mice restricted metastatic progression of colon cancer cells (*de Sousa e Melo et al., 2017*). Time-lapse live imaging analysis of metastatic cancer cells in the murine primary colon tumors revealed that both *Lgr5* positive and negative cancer cells are able to escape from the primary tumors and circulate in the bloodstream (*Fumagalli et al., 2020*). Importantly, generation or presence of *Lgr5* positive cells is necessary to form metastases, suggesting that cancer stem cells and their niche formation upon metastases seeding is a prerequisite of metastatic tumor growth.

The stem cell niche provides the so-called 'stem cell niche factors' in order to regulate proliferation and differentiation of stem cells (*McCarthy et al., 2020*; *Santos et al., 2018*). In the homeostatic condition, Paneth cells function as stem cell niche cells by providing Wnt3, EGF, Notch ligand, and Dll4 to intestinal stem cells (*Sato et al., 2011*). Paneth cell-mediated glucose conversion into lactate promotes proliferation of intestinal stem cells that lack glucose metabolism (*Rodríguez-Colman et al., 2017*). Stem cell niche factors not only maintain stem cells but also generate new stem cells by dedifferentiation from the progenitors when the original stem cells are damaged by intestinal injury or inflammation (*Tetteh et al., 2016*; *van Es et al., 2012*; *Buczacki et al., 2013*). Regeneration of stem cells by niche factors also occurs in the context of colorectal cancer (*de Sousa e Melo et al., 2017*; *Batlle and Clevers, 2017*; *Hilkens et al., 2017*). However, the underlying mechanisms of cancer stem cell niche cell development in the primary and metastatic colon tumors are largely unknown.

In normal mucosa, the balance between Wnt and Notch signaling determines the fate of intestinal stem cells differentiating to secretory or absorptive precursors (*van Es et al., 2005*; *Suzuki et al., 2005*; *Andreu et al., 2008*). *Atoh1* expression in the secretory precursors directs their differentiation to Paneth or goblet cells (*Lueschow and McElroy, 2020*). *Sox9* expression then induces Paneth cell differentiation (*Bastide et al., 2007*; *Mori–Akiyama et al., 2007*) in the small intestine. Recent findings have identified the presence of Paneth-like cells in the large intestine (*Sasaki et al., 2016*; *Wang et al., 2020*; *Qi et al., 2023*). However, the existence of Paneth or Paneth-like cells in colon cancers remains unknown. Using colon cancer organoids carrying mutations in *Apc*, *Kras*, and *Trp53* genes, we showed here that *Dkk2* KO disrupted lysozyme (LYZ) positive cell formation in colon cancer organoids as well as metastatic nodules in vivo. Single-cell RNA sequencing (scRNA-seq) analyses for mouse and human metastatic colon cancer samples have identified the existence of LYZ+ cells exhibiting Paneth cell properties including metabolic support for stem cells (*Dayton and Clevers, 2017*). Mechanistically, DKK2 protein deficiency recovered protein levels of hepatocyte nuclear factor 4 alpha (HNF4A) in colon cancer cells (*Shin et al., 2021a*). HNF4A binding on the *Sox9* promoter inhibited the formation of LYZ+ cancer cells exhibiting Paneth cell characteristics. The loss of LYZ+ cells by *Dkk2* knockout rescued mice from the liver metastasis of colorectal cancer induced by cancer organoid transplantation. Our findings suggest that DKK2 promotes LYZ+ cell formation exhibiting Paneth cell properties to develop cancer stem cell niches for outgrowth of metastasized colon cancers.

## Results

### DKK2 is indispensable for liver metastasis of colorectal cancer

Metastatic potential of *Lgr5*-expressing cells has been reported in colorectal cancer (*Li et al., 2016*). The ablation of *Lgr5*-expressing cells in primary colon tumors inhibited metastasis whereas the growth of primary tumors quickly recovered by dedifferentiation of non-stem cells into *Lgr5*-expressing stem cells (*de Sousa e Melo et al., 2017*). Moreover, Fumagalli et al. have shown that *Lgr5*-expressing cells are indispensable for the growth of metastases (*Fumagalli et al., 2020*). We have recently reported that DKK2 enhanced *Lgr5* expression in colon cancers through activation of c-Src (*Shin et al., 2021a*). Enhanced activation of c-Src has been reported in metastatic colon cancers as well (*Chen et al., 2014*). Based on these observations, we queried whether DKK2 is required for metastasis of colorectal

cancer. We developed colon cancer organoids carrying oncogenic mutations in *Apc*, *Kras,* and *Trp53* genes (AKP) (*Shin et al., 2021a*). Splenic injection of the *Dkk2* knockout AKP (KO) organoids into wild-type C57BL/6 mice resulted in significantly less liver metastasis and mortality compared to the control (AKP) organoids (*Figure 1A–D*). Quantitative gene expression analyses revealed that the levels of *Dkk2* and *Lgr5* were still decreased in KO-derived liver metastasized cells compared to AKP-derived cells (*Figure 1E*). Lgr5 protein expression and c-Src phosphorylation were reduced in KO-derived liver tumors (*Figure 1F and G*). The percentage of *Lgr5*-expressing cells with c-Src phosphorylation in KO tumor tissues was decreased about twofold compared to AKP tissues (*Figure 1H*). These data suggest the necessity of DKK2 in the growth of *Lgr5*-expressing cancer stem cells in liver metastasized nodules originated from the splenic injection of cancer organoids.

## DKK2 is required for LYZ+ cell formation in colon cancer organoids

To understand the underlying mechanisms of the DKK2-mediated increase of *Lgr5*-expressing stem cells in liver metastases, we performed RNA sequencing (RNA-seq) analysis on AKP and *Dkk2* knockout (KO) organoids. KO organoids were generated using the CRISPR technique and DKK2 reconstitution was followed using recombinant mouse DKK2 protein (*Figure 2A*). Consistent with our previous report using a colitis-induced tumor model, expression of stem cell marker genes including *Lgr5* was reduced in KO organoids (*Figure 2—figure supplement 1*; *Shin et al., 2021a*). In particular, the expression of Paneth cell marker genes such as *Lyz1*, *Lyz2*, and alpha defensins were significantly reduced in KO organoids (*Figure 2—figure supplement 1*). Paneth cells are derived from *Lgr5*-expressing stem cells by asymmetric division and maintain their *Lgr5* expression (*Buczacki et al., 2013*). These cells produce stem cell niche factors such as Wnt3A in order to facilitate stem cell proliferation and differentiation into other epithelial lineage cells (*Sato et al., 2011*). Paneth cells are also crucial for the glycolysis of intestinal stem cells that lack of glucose metabolism (*Rodríguez-Colman et al., 2017*). Our bulk RNA-seq data showed that Paneth cell-derived Wnt3a as well as *Fgfr3*, a marker of Paneth cell-induced expansion of stem cells, were reduced by *Dkk2* knockout, indicating that the formation of stem cell niche might be disrupted in KO organoids (*Figure 2—figure supplement 1*; *Vidrich et al., 2009*). Indeed, formation of LYZ+ cells was disrupted by *Dkk2* knockout during organoid culture (*Figure 2C*). LYZ+ cells were detected in AKP organoids from day 2 after plating a single or multiple cells of enzyme-digested organoids whereas those cells were less frequently observed in KO organoids. It correlates to the expression pattern of *Lgr5* in our previous report (*Shin et al., 2021a*). A significant increase of *Lgr5* expression was observed between day 2 and day 4 and peaked at day 4 to day 6 in AKP organoid culture. *Lgr5* expression was reduced about threefold in KO organoids compared to AKP organoids at day 8 (*Shin et al., 2021a*). Likewise, the markers of Paneth cells, *Lyz1* and *Lyz2,* were reduced in KO organoids at day 8 and recombinant mouse DKK2 protein treatment into KO organoids partially rescued it (*Figure 2D*). These data indicate that DKK2 is necessary for LYZ+ cell formation in colon cancer organoids that might be required for the cancer stem cell niche formation.

## LYZ+ cancer cells exhibit Paneth cell properties in both mouse and human systems

To investigate the necessity of DKK2 in the formation of LYZ+ cancer cells during colon cancer metastasis, we employed the murine in vivo model of liver metastasis shown in *Figure 1*. In order to analyze transcriptome changes by *Dkk2* knockout in each cell type of metastasized tumors, we performed scRNA-seq analysis on liver metastasized AKP and KO tumor cells. The Louvain method revealed four clusters in our dataset. A cell type was assigned to each cluster on the basis of gene expression patterns and visualized in a uniform manifold approximation and projection (UMAP) representation (*Figure 3A*). Epithelial tumor cells were sub-clustered as seven clusters and the cluster 6 has been identified as cancer cells with Paneth cell properties based on the expression of Paneth marker genes using the Panglao database (*Figure 3A*). Total of 17 cells in cluster 6 were further defined as cancer cells with Paneth cell properties (Paneth+) or goblet cell properties (Goblet+) using the relevant marker genes expression, *Lyz1*, *Muc2,* and *Atoh1* (*Figure 3—figure supplement 1*). The calculated ratio between Paneth+ cells and Goblet+ cells in KO metastasized tumors was 0.43, reduced as half compared to the ratio in the AKP control, 1. This suggests a reduction of LYZ+ cells in metastasized colon tumors by knockout a *Dkk2* gene.

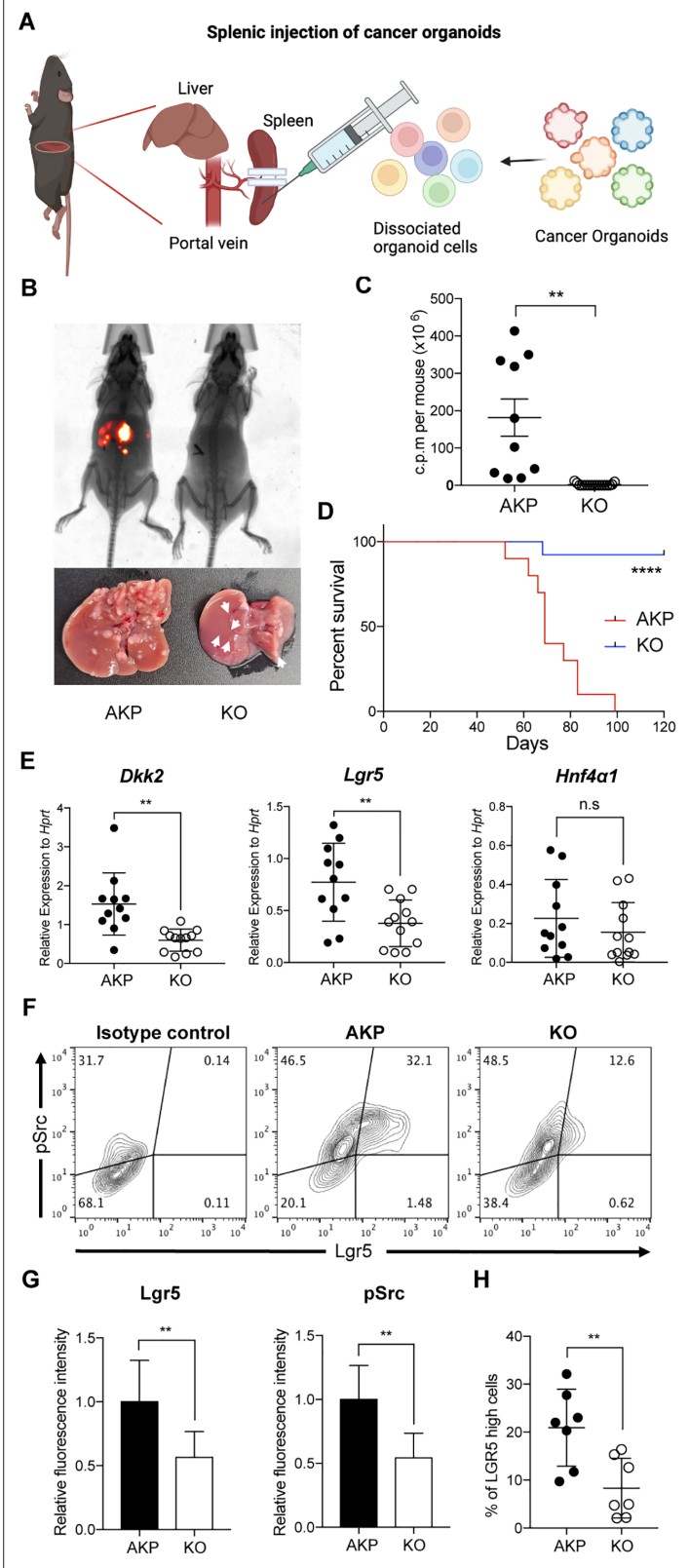

**Figure 1.** DKK2 is required for Lgr5 positive stem cells-driven liver metastasis of colorectal cancer. (**A**) Isolated cells from AKP tumor organoids expressing the *tdTomato* reporter gene were dissociated and injected into the spleen of wild-type C57BL/6 mice. 8 weeks after the injection, metastatic tumor growth was measured by in vivo imaging analysis. (**B**) Representative pictures of in vivo imaging analysis. Ctrl: AKP control organoids transduced

*Figure 1 continued on next page*

*Figure 1 continued*

with scrambled guide RNA, KO: *Dkk2* knockout AKP organoids. (**C–D**) Statistic analysis of liver metastasis (**C**) and survival (**D**). Ctrl (n=10), KO (n=15), n represents the number of mice. c.p.m. in (**C**): count per minute. (**E**) Quantitative gene expression analysis of *Dkk2*, *Lgr5*, and *Hnf4α1* in liver metastasized colon cancer cells. Ctrl (n=11), KO (n=12), n represents the number of isolated cancer nodules with five mice per group. (**F**) Representative flow cytometry analysis data of c-Src phosphorylation (pSrc) and *Lgr5* expression in metastasized colon cancer cells. (**G**) Statistic analysis of *Lgr5* expression and pSrc in (**E**). The average of mean fluorescence intensity in control samples was set as 1 and relative fluorescence intensity was calculated. (**H**) Statistic analysis of the percentile of Lgr5 high (Lgr5 and pSrc double positive) cells in metastasized colon cancer in (**F**). Each *symbol* represents an individually isolated cancer nodule. n.s.=not significant, **p<0.01, ****p<0.0001; two-tailed Welch's t-test (**C, E, G, H**). Error bars indicate mean ± s.d. Log-rank test (**D**). Results are representative of three independent experiments.

To analyze the consequence of the loss of LYZ+ cells by *Dkk2* knockout in cancer stem cell niche formation, we enriched *Lgr5* positive cells that consist of stem cells, stem cell niche cells, and transit amplifying cells in the homeostatic condition (*Figure 3—figure supplement 2*; *Sato et al., 2009*). Elevated expression of stem cell marker genes represents cancer stemness in *Lgr5* positive cells

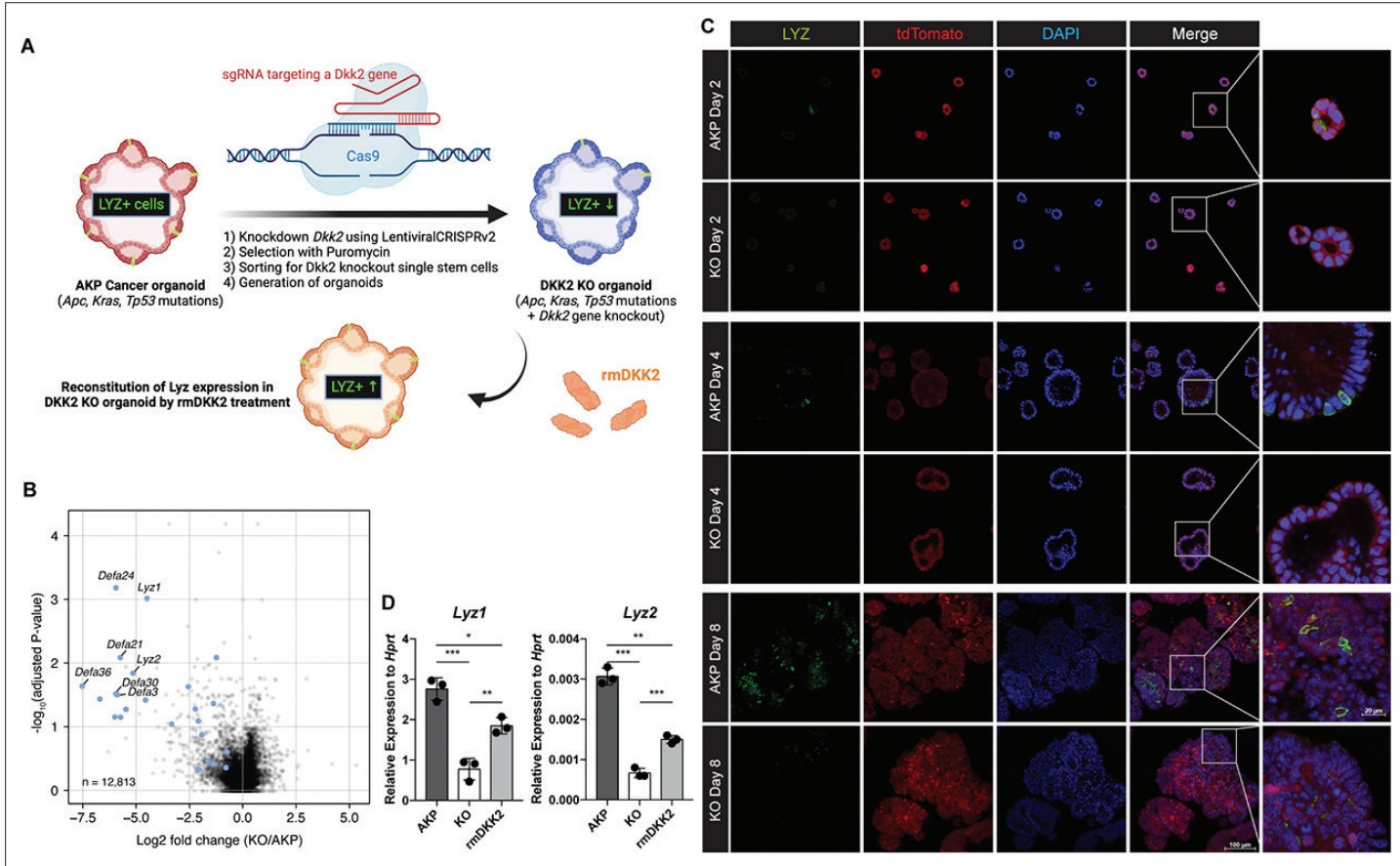

**Figure 2.** DKK2 is indispensable for the generation of cancer cells with Paneth cell properties in colon cancer organoids. (**A**) A schematic diagram of the generation of KO organoids using CRISPR technique and the reconstitution of DKK2 in organoids by recombinant mouse DKK2 protein (rmDKK2) treatment. Lysozyme (LYZ) expression is highlighted for the following. (**B**) A volcano plot of RNA sequencing (RNA-seq) analysis comparing KO versus AKP organoids. Paneth cell marker genes are highlighted as blue circles (AKP = 3 and KO = 5 biological replicates were analyzed). (**C**) Confocal microscopy analysis of LYZ positive cells in AKP or KO organoids in a time-dependent manner using anti-LYZ antibody. (**D**) Quantitative real-time PCR analyses of *Lyz1* and *Lyz2* in 8 days cultured colon cancer organoids. KO organoids were cultured in the presence of 1 µg/ml of recombinant mouse DKK2 protein. *p<0.05, **p<0.01, ***p<0.001; two-tailed Welch's t-test. Error bars indicate mean ± s.d. Results are representative of three biological replicates.

The online version of this article includes the following figure supplement(s) for figure 2:

**Figure supplement 1.** Bulk RNA sequencing (RNA-seq) analysis of *Dkk2* knockout cancer organoids.

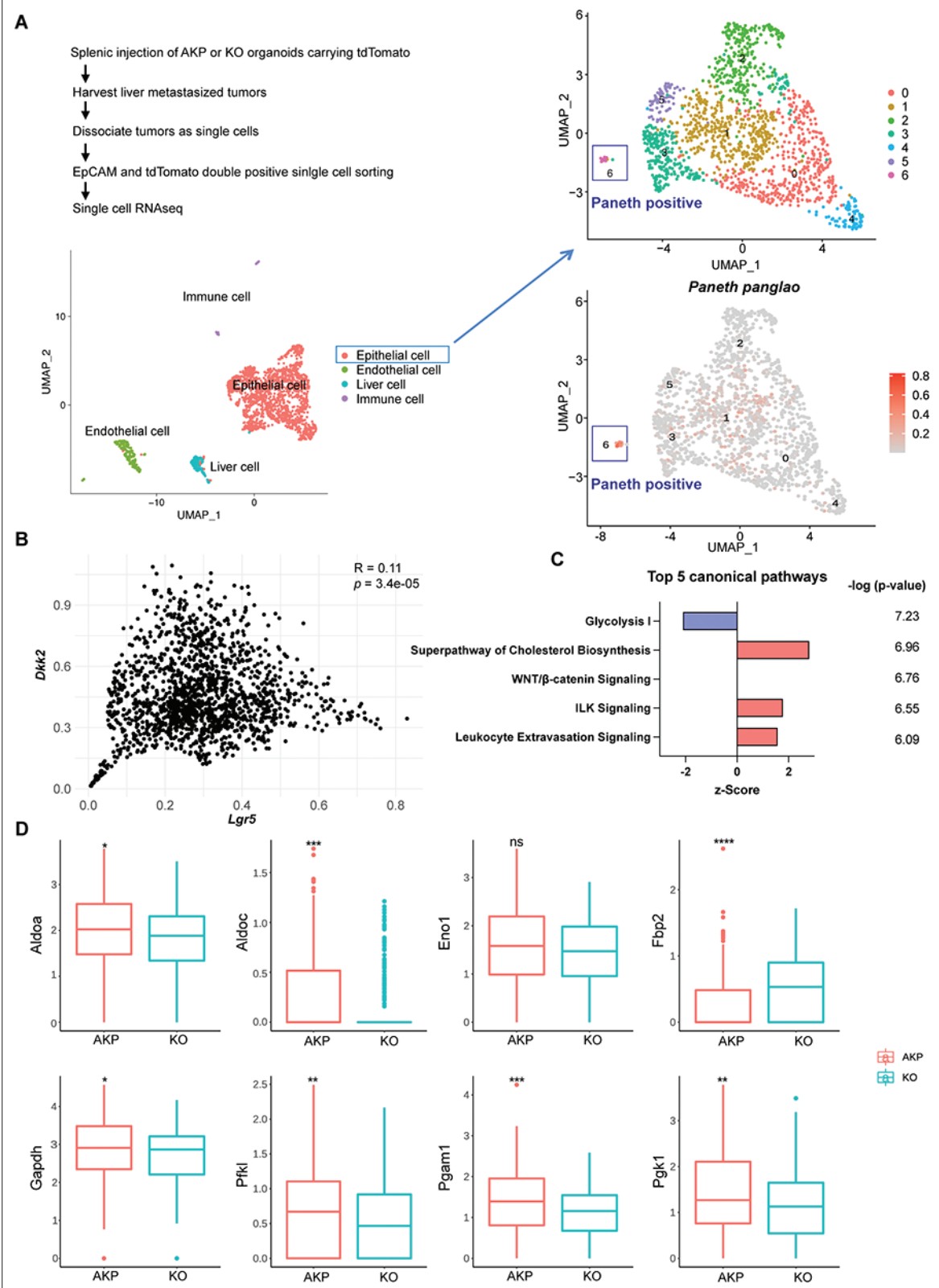

**Figure 3.** Cancer cells harboring Paneth cell properties were reduced in *Dkk2* knockout metastasized colon cancer tissues in mice. Control AKP or KO colon cancer organoids were transplanted via splenic vein as described in *Figure 1*. 3 weeks after transplantation, mice were sacrificed to analyze metastatic tumor growth in liver. (**A**) Single-cell RNA sequencing analysis (scRNA-seq) of liver metastasized colon cancer tissues. The uniform manifold approximation and projection (UMAP) plot clustered epithelial, endothelial, liver, and immune cells in metastasized cancers based on transcriptome

*Figure 3 continued on next page*

*Figure 3 continued*

analysis. The cancer epithelial cell cluster was sub-clustered to identify cells with Paneth cell properties (cluster 6, Paneth positive). (**B**) The correlation between *Dkk2* and *Lgr5* expression in the cluster of epithelial cells by Pearson r test. (**C**) Ingenuity pathway analysis (IPA)-suggested top 5 canonical pathways of the scRNA-seq data of *Lgr5* positive epithelial cells in KO compared to AKP. z-Scores indicate activation or inhibition of the suggested pathways. The significance values for the pathways are calculated by the right-tailed Fisher's exact test. (**D**) Box plots show expressions of the genes involved in the glycolysis I pathway in (**C**). ns = not significant, *p<0.05, **p<0.01, ***p<0.001, ****p<0.0001; Wilcoxon signed-rank test.

The online version of this article includes the following figure supplement(s) for figure 3:

**Figure supplement 1.** Identification of the cluster of Paneth-like cells in the single-cell RNA sequencing (scRNA-seq) data of liver metastasized murine colon cancer cells.

**Figure supplement 2.** Reduced expression of *Noggin (Nog)* and reversed expression of *Bmp4* in the single-cell RNA sequencing (scRNA-seq) of *Dkk2* knockout metastasized cancer cells.

compared to *Lgr5* negative cells. The UMAP plot of *Lgr5* positive cells displayed two clusters. Interestingly, expression of Noggin (*Nog*), one of the stem cell niche factors, was detected mostly in cluster 1 where only one cell was counted in KO. Noggin is an inhibitor of BMP signaling, which restricts stemness of *Lgr5* positive stem cells in the gut (*Haramis et al., 2004*; *Qi et al., 2017*). Following the reduction of *Nog*, *Bmp4* expression was reversed in KO tumor epithelial cells in the scRNA-seq data (*Figure 3—figure supplement 2*). This has been confirmed by quantitative gene expression analysis of *Bmp4* in KO organoids. The correlative expression between *Dkk2* and the stem cell marker gene, *Lgr5*, has been shown in the scRNA-seq data (*Figure 3B*). Furthermore, pathways enrichment analysis of the scRNA-seq data using ingenuity pathway analysis (IPA) suggested that the glycolysis I pathway was significantly inhibited in KO *Lgr5* positive tumor epithelial cells (*Figure 3C*). Expression of glycolysis-related genes such as *Aldoa*, *Aldoc*, and *Fbp2* was reduced by *Dkk2* knockout in *Lgr5* positive cells (*Figure 3D*; *Cascone et al., 2018*). Previous studies have defined that Paneth cells participate in the regulation of energy metabolism in intestinal stem cells (*Rodríguez-Colman et al., 2017*; *Yilmaz et al., 2012*). Since stem cells are not able to generate lactate, Paneth cells instead process glucose in order to provide lactate to stem cells (*Rodríguez-Colman et al., 2017*). Therefore, reduction of the glycolysis I pathway in the absence of DKK2 indicates the impaired stem cell niche formation by the loss of LYZ+ cells carrying Paneth cell properties.

Paneth cells constitute the stem cell niche in the small intestine (*Lueschow and McElroy, 2020*). Recent studies have identified the presence of Paneth-like cells in the colon (*Wang et al., 2020*; *Qi et al., 2023*). However, the presence of Paneth cells or Paneth-like cells within the tumor microenvironment is elusive. To discern the existence of cancer cells exhibiting Paneth cell properties in humans, we conducted an analysis of scRNA-seq data obtained from colorectal cancer patients (*Joanito et al., 2022*). Our analysis focused on CD45-EpCAM+ epithelial cells, excluding hematopoietic cells. UMAP plotted 31 clusters of single cells derived from normal tissue, primary tumor, and metastasis samples (*Figure 4A and B*). Employing LYZ expression and Paneth cell module scores (Pangalo database) as criteria, we have identified LYZ+ cancer cells harboring Paneth cell properties across five clusters (*Figure 4C–H*, *Figure 4—figure supplements 1 and 2*). LYZ+ cancer cell population is distinct in metastasis samples (*Figure 4E and F*). LYZ expression over 2.0 (third quantile, round 1) and Paneth module scores over 0.2 (third quantile, round 2 to avoid getting 0) single cells were identified as LYZ+ cancer cells harboring Paneth cell properties, 1%, 13%, and 24% in normal, primary tumor, and metastasis samples, respectively. This delineates the distribution of LYZ+ cancer cells with Paneth cell characteristics in different stages of colorectal cancer progression.

To unravel the functional significance of LYZ+ cancer cells, we performed gene set enrichment analysis (GSEA) for the scRNA-seq data obtained from colon cancer patients. In correlation with our murine scRNA-seq data IPA results, glycolysis emerged as the second-ranked enriched pathway in LYZ+ cancer cells compared to LYZ- cancer cells (*Figure 5A*). Notably, key players in glycolysis such as *ALDOB* and *KDELR3* genes are increased in LYZ+ cancer cells (*Figure 5B*). Also, the hallmark of protein secretion was the top-ranked enriched pathway, which is essential for Paneth cell activity including secretion of anti-microbial peptides (*Figure 5A*). These results suggest the existence of a distinct population of LYZ+ colon cancer cells endowed with Paneth cell properties. Collectively, DKK2 is required for the generation of LYZ+ cells to form the cancer stem cell niche, a prerequisite for the outgrowth of metastasized colorectal cancer cells.

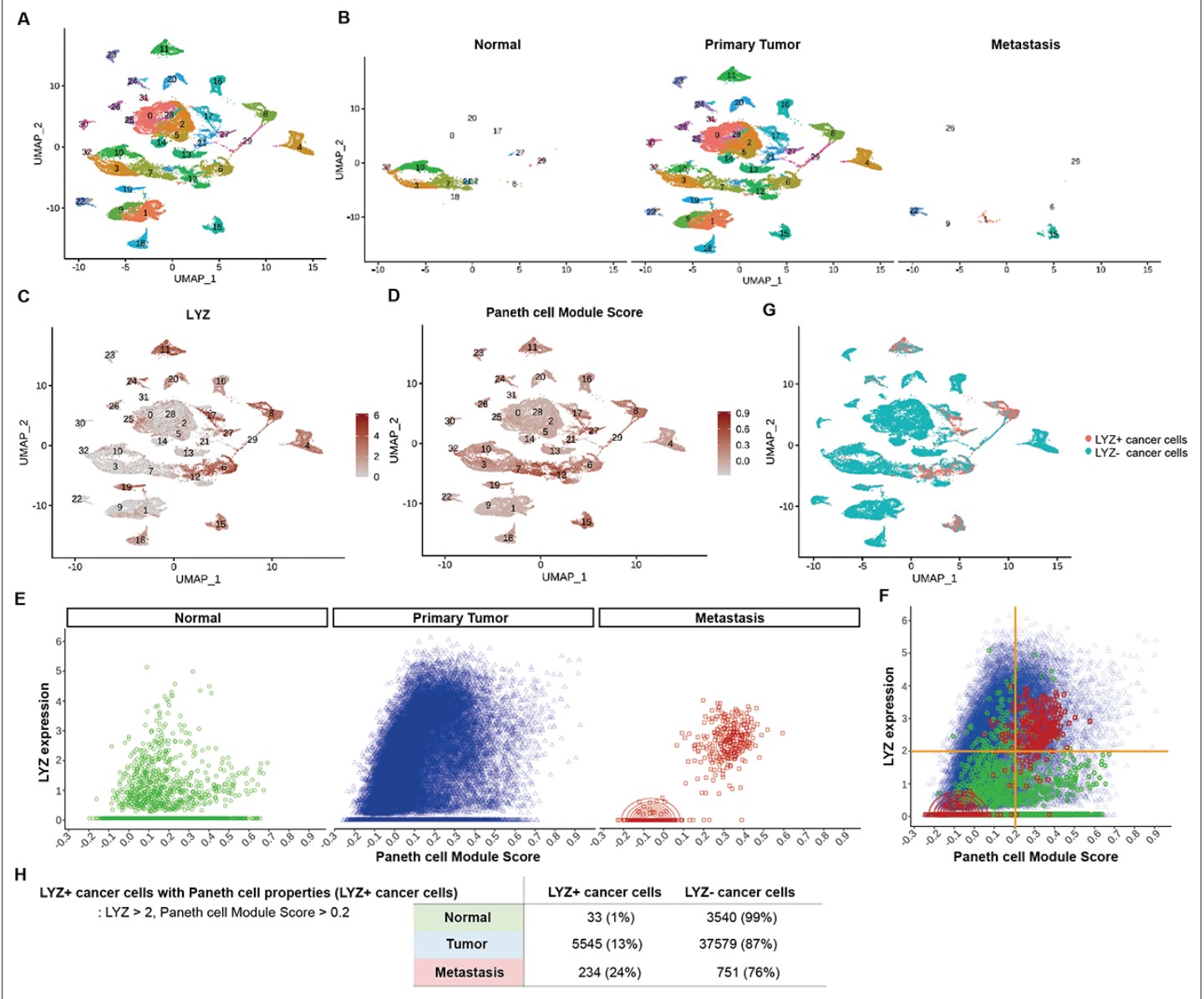

**Figure 4.** Identification of colon cancer cells harboring Paneth cell properties in humans. Published colorectal cancer patients' single-cell RNA sequencing (scRNA-seq) data was analyzed to identify the presence of cancer cells with Paneth cell properties (*Joanito et al., 2022*). (**A**) The uniform manifold approximation and projection (UMAP) plot of total 31 clusters. (**B**) Normal cells, primary tumor cells, and liver metastasized cells (metastasis) are shown in the UMAP plot clusters. (**C–D**) Expression levels of lysozyme (LYZ) and Paneth cell module scores are displayed in the UMPA plot. (**G**) Based on the analysis in (**C**) and (**D**), cancer cells harboring Paneth cell properties are indicated as red dots (LYZ+ cancer cells). (**E**) LYZ expression and Paneth cell module scores of single cells in normal, primary tumor, and metastasis samples are presented by dot plots. (**F**) Dot plots in (**E**) are overlayed. (**H**) The percentiles of cancer cells with Paneth cell properties (LYZ+ cancer cells) are shown.

The online version of this article includes the following figure supplement(s) for figure 4:

**Figure supplement 1.** Paneth cell markers expression in colorectal cancer patients' single-cell RNA sequencing (scRNA-seq) data.

**Figure supplement 2.** Analyses of the regulon activity of various transcription factors in lysozyme positive (LYZ+) colon cancer cells in human colon cancer single-cell RNA sequencing (scRNA-seq) data.

## HNF4A mediates the formation of LYZ+ colon cancer cells by DKK2

IPA of the human scRNA-seq data from primary tumor and metastasis samples suggested HNF4A as an upstream regulator of LYZ+ cancer cells compared to LYZ- cancer cells (*Figure 5C*). In line with this, IPA of the bulk RNA-seq data using mouse colon cancer organoids suggested that HNF4A is activated in KO organoids (*Figure 6A*). This is consistent with our previous report that DKK2 enhanced

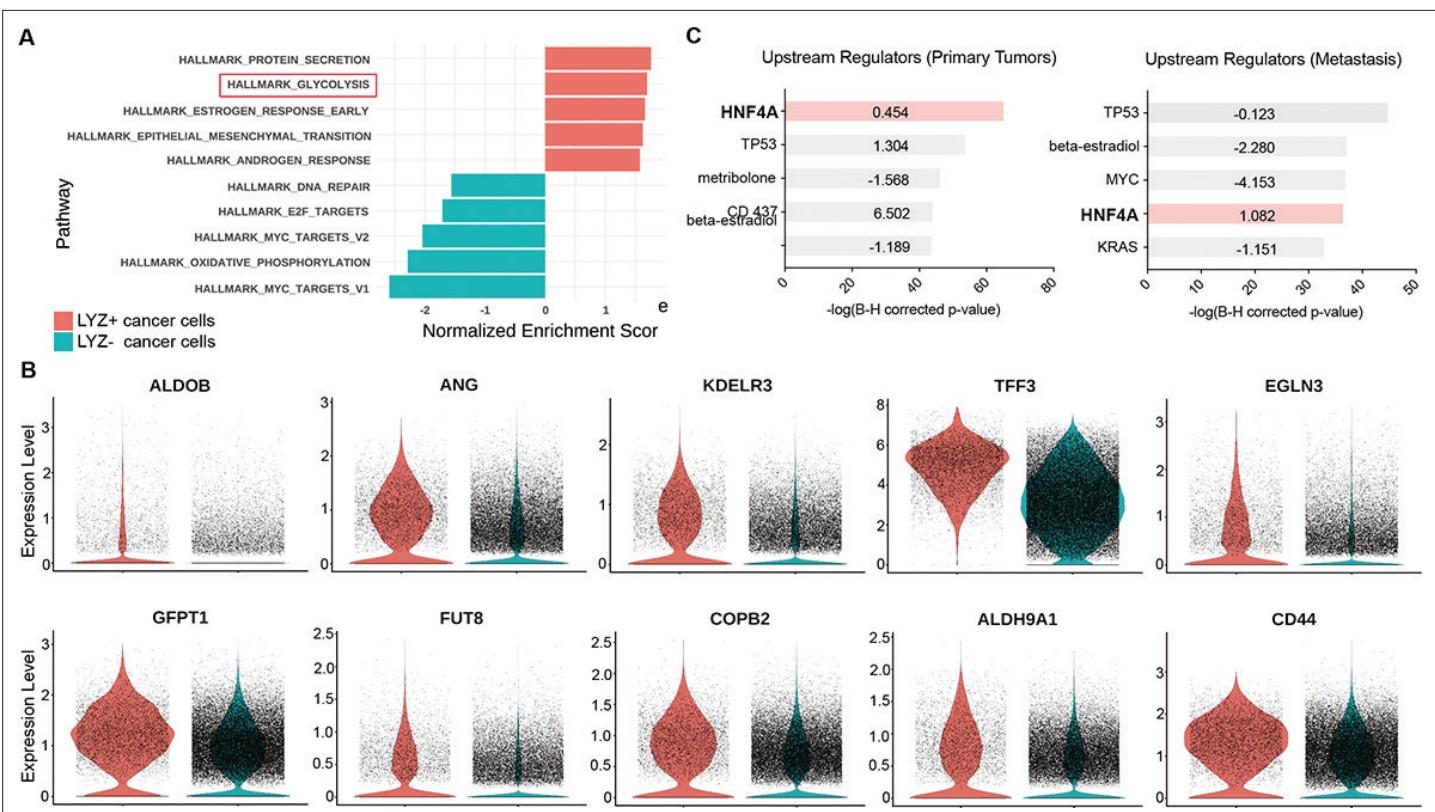

**Figure 5.** Colon cancer cells with Paneth cell properties contribute to glycolysis. Colorectal cancer patients' single-cell RNA sequencing (scRNA-seq) data were further analyzed by gene set enrichment analysis (GSEA) and ingenuity pathway analysis (IPA). (**A**) GSEA of cancer cells with Paneth cell properties (lysozyme [LYZ]+ cancer cells) compared to all other cancer cells (LYZ- cancer cells), shown in *Figure 4* (**G**). (**B**) Representative gene expressions in the hallmark of glycolysis pathway are shown. (**C**) Upstream regulators suggested by IPA in primary tumor and metastasis are presented. Activation z-scores are indicated in the bar. IPA predicted activation or inhibition of the upstream regulators are colored as red and cyan, respectively.

*Lgr5* expression in colitis-induced cancer cells via HNF4α1, an isoform of HNF4A (*Shin et al., 2021a*). Significant enrichment of the HNF4A binding motif was also detected in the open chromatin regions of KO organoids compared to AKP controls determined by assay for transposase-associated chromatin using sequencing (ATAC-seq) (*Figure 6B*). In comparison with normal colonic organoids, the HNF4A motif was listed as the most reduced motif in AKP indicating significant reduction of HNF4A-regulated transcription whereas it might be rescued by *Dkk2* knockout (*Figure 6C*). Notably, HNF4A was the only molecule suggested by all of these multi-omics data analyses in both mouse and human. These results suggest HNF4A as a key transcription factor in downstream signaling of DKK2 in colon cancer cells.

HNF4A is necessary for maturation of fetal intestine and barrier functions of intestinal epithelium in the homeostatic condition (*Chen et al., 2019*; *Cattin et al., 2009*). Two *Hnf4a* isoforms—*Hnf4a1* and *Hnf4a7*— are involved in intestinal homeostasis and regeneration (*Chellappa et al., 2016*). Intestinal inflammation induces expression of the HNF4α1 isoform in the crypt epithelial cells to regenerate the epithelium whereas loss of the HNF4α1 isoform occurs in colitis-induced tumor cells (*Chellappa et al., 2016*). Loss of the HNF4α1 isoform was also observed in AKP organoids while *Dkk2* knockout restored the presence of HNF4α1 protein in the nucleus of AKP organoid cells (*Shin et al., 2021a*). This allowed us to investigate the binding regions of HNF4α1 in colon cancer cells using KO organoids. Chromatin immunoprecipitation followed by sequencing (ChIP-seq) analysis using the ChIP-seq grade anti-HNF4A antibody revealed that HNF4A binds to the promoter region of *Sox9* in KO organoid cells (*Figure 6D*). Sox9 is a transcription factor required for the secretory lineage of epithelial cell differentiation including Paneth cells (*Mori–Akiyama et al., 2007*). Quantitative gene expression analysis showed that *Dkk2* knockout reduced *Sox9* expression in AKP organoids, which was rescued by recombinant DKK2 protein treatment (*Figure 6E*). Reduced *Sox9* expression in KO

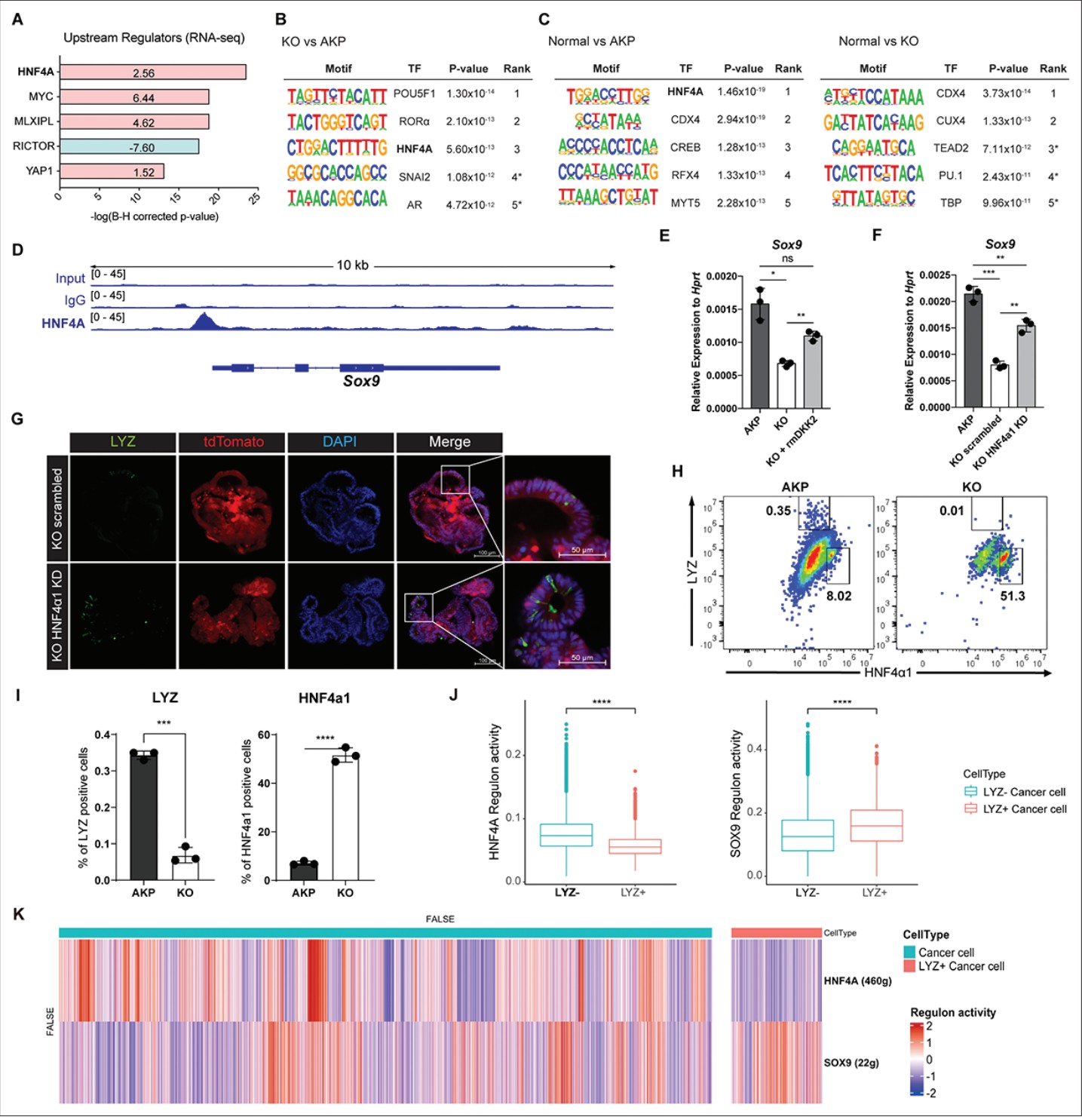

**Figure 6.** DKK2-driven reduction of HNF4α1 protein in murine colon cancer organoids promotes cancer cells with Paneth cell properties. (**A**) Ingenuity pathway analysis (IPA)-suggested upstream regulators of the RNA sequencing (RNA-seq) data shown in ***Figure 2A***. z-Scores are presented in the bar. IPA predicted activation or inhibition of the upstream regulators are colored as red and cyan, respectively. (**B–C**) List of top 5 transcriptional factors (TF) in the motif enrichment analysis of assay for transposase-associated chromatin using sequencing (ATAC-seq) data comparing normal colonic organoids, AKP, and KO cancer organoids. * indicates possible false positive. (**D**) Chromatin immunoprecipitation sequencing (ChIP-seq) analysis of knockout (KO) organoids using an anti-HNF4A antibody. (**E**) Quantitative expression of *Sox9* in colon cancer organoids in the presence or absence of DKK2 (rmDKK2: recombinant mouse DKK2 protein added in organoid culture). (**F**) Analysis of Sox9 expression after knockdown *HNF4α1* in KO colon cancer organoids (KO HNF4α1 KD). ns = not significant, *p<0.05, **p<0.01, ***p<0.001; two-tailed Welch's t-test. Error bars indicate mean ± s.d. (**G**) Representative

*Figure 6 continued on next page*

*Figure 6 continued*

images of confocal microscopy analysis of Lyz-stained cancer cells with Paneth cell properties in DKK2 KO HNF4α1 KD organoids. (**H–I**) Control AKP or KO colon cancer organoids were transplanted via splenic vein as described in *Figure 1*. 3 weeks after transplantation, mice were sacrificed to analyze metastatic tumor growth in liver. Quantification of cancer cells with Paneth cell properties in metastasized tumor nodules by flow cytometry for lysozyme (LYZ) and HNF4α1. Tumor cells were initially gated by the tdTomato reporter expression. Representative images of flow cytometry are shown (**H**). Statistic analyses of the percentiles of LYZ positive cells (% of upper left) and HNF4α1 positive cells (% of lower right) in tumor nodules (**I**). **p<0.01, ****p<0.0001; two-tailed Welch's t-test. Error bars indicate mean ± s.d. Three mice were tested per group. Data are representative of two independent experiments. (**J– K**) Reduced HNF4A regulation activity with enhanced SOX9 regulation activity in LYZ+ cancer cells in human colorectal cancer scRNA-seq data. Box plots represent the regulon activity of HNF4A and SOX9 in LYZ+ cancer cells. ****p<0.0001; Wilcoxon signed-rank test (**J**). z-Scaled regulon activities of HNF4A and SOX9 in human colon cancer cells are displayed by heatmap (**K**).

organoids was also reversed by HNF4α1 knockdown (*Figure 6F*). These data indicate that DKK2-driven loss of HNF4α1 protein leads to *Sox9* expression followed by formation of LYZ+ colon cancer cells. Confocal microscopy analysis confirmed that HNF4α1 knockdown rescued the presence of LYZ+ cancer cells in KO organoids (*Figure 6G*). In order to quantify LYZ+ cancer cells in the mouse model of liver metastasis, tdTomato positive cells in the liver were identified as the implanted organoid-derived cancer cells then, LYZ+ cells were calculated (*Figure 6H*). The percentile of LYZ+ cells in total cancer cells was decreased about threefold in *Dkk2* knockout metastasized cells (KO) compared to the AKP control cells in liver (*Figure 6I*). Simultaneously, HNF4α1 positive cells were tripled to about 60% in KO (*Figure 6H and I*). These findings suggest that DKK2 is indispensable for the generation of LYZ+ cancer cells in liver metastasized nodules by reducing protein levels of HNF4α1. Further analyses for human colorectal cancer scRNA-seq data revealed that HNF4A regulation activity is reduced in LYZ+ human colon cancer cells compared to the LYZ- cancer cells while SOX9 regulation activity is significantly increased (*Figure 6J and K*) suggesting the key roles of HNF4A and SOX9 in LYZ+ cancer cell formation in human. Taken together, DKK2-driven loss of HNF4α1 protein enhances Sox9 expression in colon cancer cells to generate LYZ+ cells with Paneth cell properties.

## Discussion

We have shown that DKK2 is indispensable for metastatic tumor growth of colorectal cancer in the murine model developed by transplantation of colon cancer organoids carrying mutation in the *Apc*, *Kras*, and *Trp53* genes. Bulk RNA-seq analysis of DKK2-deficient colon cancer organoids showed reduction of the marker genes of Paneth cells, which play a role in the stem cell niche development. scRNA-seq analyses for mouse and human metastatic colon cancers have identified the existence of LYZ+ cancer cells harboring Paneth cell properties, in particular, glycolysis for stem cells. In the absence of DKK2, both pathway enrichment analysis of the RNA-seq and ATAC-seq data suggested that HNF4A is activated. This is consistent with our previous report in the murine model of colitis-induced tumorigenesis (*Shin et al., 2021a*). ChIP-seq analysis using an anti-HNF4A antibody revealed that HNF4A directly binds to the promoter region of *Sox9*, a transcriptional factor required for epithelial differentiation into Paneth cells (*Bastide et al., 2007*; *Mori–Akiyama et al., 2007*). Notably, DKK2 induces the loss of HNF4α1 isoform in colon cancer cells followed by elevation of *Sox9* expression lead to the formation of LYZ+ cancer cells exhibiting Paneth cell properties. DKK2-deficient colon cancer cells possessed high levels of HNF4α1 protein and failed to generate LYZ+ cancer cells in vitro and in vivo. DKK2 deficiency resulted in reduced glycolysis in mouse liver metastasized colon cancer cells. These results demonstrate the significant reduction of liver metastasis in DKK2 KO colon cancer cells infused mice.

*Fumagalli et al., 2020*, have recently shown that the generation of *Lgr5* positive cells from the metastasized *Lgr5* negative cancer cells is necessary for their outgrowth using in vivo time-lapse imaging and organoid culture of metastatic cancer cells that escaped from the primary colon cancer tissues (*Fumagalli et al., 2020*). This finding implies the necessity of both cancer stem cells and their niche for metastatic outgrowth. Indeed, Paneth cell-derived stem cell niche factors such as Wnt3, EGF, and Dll4 are essential for the maintenance of stem cells (*Sato et al., 2011*). Dietary-induced mouse sporadic intestinal cancer showed elevation of Paneth cell marker genes in the intestinal epithelium (*Wang et al., 2011*). IDO1+ Paneth cells contribute to the immune evasion of colon cancer (*Pflügler et al., 2020*). Depletion of Paneth cells or Paneth cell-derived Wnt3 in *Apc^Min* mice impaired intestinal adenoma formation (*Chen et al., 2021*). The plasticity of Paneth cells allows dedifferentiation of

Paneth cells into intestinal stem cells (**Buczacki et al., 2013**; **Clevers, 2013**). These reports indicate the necessity of the Paneth cells during the development of colon cancer. However, the existence of Paneth cells in large intestine or tumor tissues remains unclear. In 2016, Reg4+ secretory cells have been suggested as Paneth-like cells in the colon (**Sasaki et al., 2016**). Single-cell transcriptome analysis showed the presence of Paneth-like cells in human colon (**Wang et al., 2020**). Recent study further supported the presence of Paneth-like cells within the colonic stem cell niche during bifidobacterium-induced intestinal stem cell regeneration (**Qi et al., 2023**). Our study suggests the presence of LYZ+ cells exhibiting Paneth cell properties in the context of colon cancer and metastasis. This extends our knowledge of colorectal cancer progression that DKK2 is required for the generation of LYZ+ cells forming cancer stem cell niches.

Further investigation will be needed to characterize LYZ+ cancer cells with Paneth cell properties observed in colon cancers carrying mutations in *Apc*, *Kras,* and *Trp53* genes compared to normal Paneth cells in order to identify their roles in cancer stem cell niche development. Recent studies have reported stem cell niche remodeling in the primary tumor in the presence of oncogenic mutations. Cancer stem cells outcompete their neighboring normal stem cells during intestinal tumorigenesis, which is known as clonal fixation (**Vermeulen et al., 2013**). *Apc*-mutant cancer cells secrete several Wnt antagonists such as NOTUM to inhibit proliferation of normal intestinal stem cells and facilitate their differentiation (**Flanagan et al., 2021**). Other oncogenic mutations in *Kras* or *Pi3kca* genes lead to paracrine secretion of BMP ligands, remodeling the stem cell niche that is detrimental for maintaining normal stem cells (**Yum et al., 2021**). These changes are beneficial for the outgrowth of cancer stem cells in the crypt where the stem cell niche had been developed prior to tumorigenesis. In the context of metastasis, the formation of stem cell niches is required upon metastases seeding in the liver. We have shown that DKK2 is required for the formation of LYZ+ cancer cells carrying Paneth cell properties. DKK2 expression is induced by *Apc* knockout and further increased by additional oncogenic mutations in *Kras* and *Trp53* genes (**Shin et al., 2021a**). Characterization of DKK2-induced LYZ+ cancer cells will provide better understanding of the biology of cancer stem cell niches, particularly in liver metastases of colorectal cancer.

DKK2 expression in colon cancer promotes LYZ+ cancer cell generation through the regulation of HNF4α1, which suppresses expression of *Sox9*, the transcription factor for Paneth cell differentiation. It has been recently reported that HNF4α2 isoform acts as an upstream regulator of Wnt3 and Paneth cell differentiation implying the roles of HNF4A for Paneth cell differentiation in the homeostatic condition (**Jones et al., 2023**). Using *Villin^Cre^-Apc^floxed^* mice, Suzuki et al. have confirmed that HNF4A proteins are completely absent in cancer cells with high β-catenin activity along with increased *Sox9* expression (**Suzuki et al., 2023**). *Sox9* regulates cell proliferation and is required for Paneth cell differentiation (**Bastide et al., 2007**; **Mori–Akiyama et al., 2007**). Bi-allelic inactivation of the *Apc* gene induces *Sox9* expression in colon cancer (**Feng et al., 2013**). Likewise, *Apc* inactivation induces *Dkk2* expression (**Shin et al., 2021a**). Here, we showed that DKK2-mediated HNF4α1 protein degradation enhanced *Sox9* expression in colon cancer (**Figure 7**). Notably, *Apc* mutation-driven *Sox9* expression blocks intestinal differentiation and activates stem cell-like program (**Feng et al., 2013**). This is consistent with our previous report that DKK2 enhances *Lgr5* expression in colon cancer (**Shin et al., 2021a**). In sum, our findings suggest that a Wnt ligand, DKK2, is an upstream regulator of SOX9 expression for enhanced stem cell activity via the formation of LYZ+ cells with Paneth cell characteristics in colorectal cancers.

In this study, we have shown that DKK2 promotes liver metastasis of colon cancer. DKK2 is required for the formation of LYZ+ colon cancer cells exhibiting Paneth cell properties such as glycolysis for stem cells. Glycolysis in *Lgr5*-expressing cancer cells was reduced along with the loss of LYZ+ cells by *Dkk2* knockout. As a result, metastatic growth of colon cancer cells was significantly inhibited in vivo. These findings suggest the necessity of DKK2 in the development of cancer stem cell niches for developing metastases. Mechanistically, DKK2-mediated degradation of HNF4α1 protein elevated *Sox9* expression followed by LYZ+ cell formation. Given that DKK2 expression is intestinal tumor specific, inhibition of DKK2 may have promising therapeutic outcomes in the treatment of metastatic colon cancer.

## Materials and methods
### Animals
All animal studies were conducted with the approval of the Institutional Animal Care and Use Committee at the Yale University School of Medicine (protocol #2020-11211). In this research, 8- to

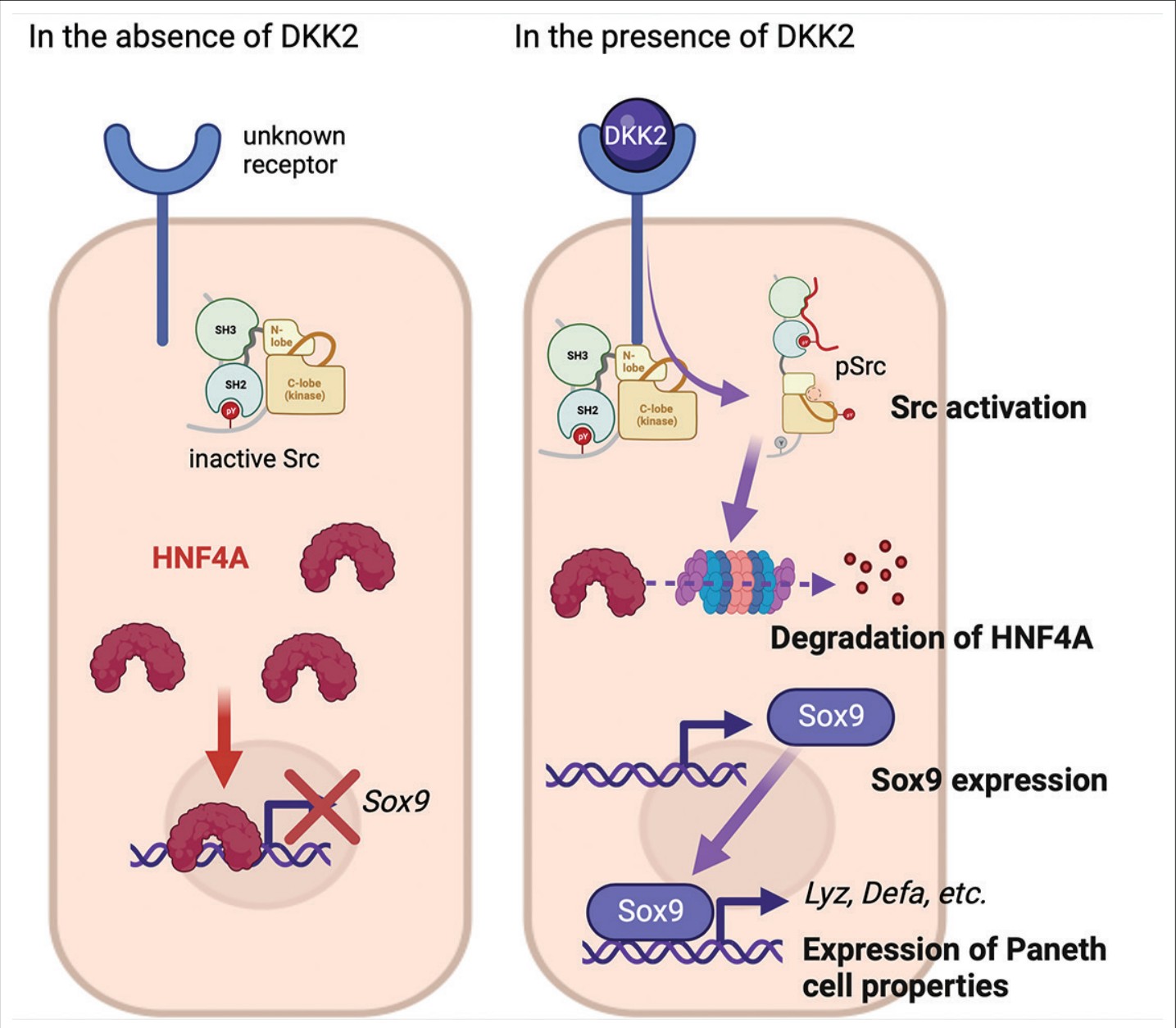

**Figure 7.** The suggested mechanism of DKK2 in the formation of colon cancer cells with Paneth cell properties. In the absence of DKK2, Sox9 expression is inhibited by HNF4A. Our previous report has shown that DKK2 activates Src followed by degradation of HNF4A protein (*Shin et al., 2021a*). HNF4A deficiency leads to Sox9 expression in colon cancer cells that induces Paneth cell properties including the expression of lysozymes and defensins. Formation of cancer cells with Paneth cell properties by DKK2 contributes to the outgrowth of metastasized colon cancer cells in the liver.

12-week-old C57BL/6 mice were utilized for colon cancer organoid transplantation via splenic injection, to develop liver metastasis. All procedures were performed in accordance with ethical guidelines to ensure the welfare and humane treatment of the animals involved in the study.

## Splenic injection

Generation of AKP, *Dkk2* knockout AKP, *Hnf4a1* knockdown, and *Dkk2* knockout AKP organoids and their culture protocols have been published in the previous study (*Shin et al., 2021a*). We used the previously reported splenic injection methodology to develop organoid-derived liver metastasis in mice (*Shin et al., 2021b*). Cultured organoids were prepared as single cells following 10–30 min of digestion in TrypLE solution (Thermo, 12605010) at 37°C, followed by centrifugation with advanced

DMEM-F12 (Thermo, 12634010). The isolated single cells were re-suspended with PBS as 300,000 cells per 100 µl of PBS. Surgical instruments were sterilized by steam autoclave (250°F, 15 p.s.i. for 30 min). 8- to 12-week-old C57BL/6 mice were anesthetized with ketamine (100 mg/kg) plus xylazine (10 mg/kg) via intraperitoneal injection. Puralube Vet ointment was applied to the eyes to prevent drying while surgery. Spleen was divided by two Horizon medium size ligating clips (Teleflex, 002200) in the center of the spleen and one hemi-spleen was returned into the peritoneum. Using a 26½ Gauge needle, 100–200 µl volume of cells was injected underneath the splenic capsule over the course of 30 s. A small Horizon clip (Teleflex, 001200) was applied to ligate the pancreas and splenic vessels and the injected hemi-spleen was removed. The incision was closed with a 5-0 running stitch using Vicryl (Ethicon, J493G) and 2–3 skin clips. Meloxicam (0.3 mg/kg) was injected subcutaneously to alleviate post-surgical pain at 24–72 hr post-surgery.

### In vivo imaging

Noninvasive in vivo imaging was performed using a Microfocus X-ray imaging source (Kodak). Mice were anesthetized by isoflurane inhalation and their abdominal hair was shaved. Tandem repeat Tomato (tdTomato) fluorescence in liver metastatic tumors was recorded for 30 s. tdTomato signals were statistically analyzed using the Bruker image analysis system. X-ray image was captured by 10 s exposure and merged with the tdTomato image.

### Tumor dissociation

Liver metastasized tumors were collected and dissociated in the digestion buffer containing 200 U/ml of type IV collagenase (Worthington, LS004188), 125 µg/ml of type II dispase (Sigma, D4693), 2.5% fetal bovine serum, 1x penicillin/streptomycin in DMEM at 37°C for 30 min with agitation. Isolated cells were centrifuged at 400×*g* and 4°C for 5 min, then re-suspended in PBS and filtered through the 40 µm pore size strainer.

### Flow cytometry

Dissociated metastatic tumor cells were fixed and permeabilized with the intracellular fixation and permeabilization buffer set (eBioscience, 88-8824-00) following the manufacturer's protocols. Those cells were stained with DyLight488-conjugated anti-Lgr5 antibody (Clone: OTIA2, Origene, TA400002), anti-phospho-Src (Ty416) antibody (Clone: 9A6, MilliporeSigma, 05677) and Brilliant Violet 421 (BV421) rat anti-mouse IgG3 antibody (Clone: R40-82, BD Horizon, 565808) in *Figure 1E*. In *Figure 4A*, tumor cells were stained with anti-LYZ antibody (Clone: Poly28600, BioLegend, 8600001), APC-conjugated goat anti-rabbit-IgG polyclonal antibody (Invitrogen, A10931), anti-HNF4a1 antibody (Clone: K9218, R&D Systems, PP-K9218-00), and Alexa Fluor 488-conjugated goat anti-mouse IgG2a secondary antibody (Thermo Fisher Scientific, AB_2535771). Flow cytometry was performed using the Stratedigm-13.

### RNA isolation

Total RNA of organoids and liver metastasized tissues were extracted using TRIzol reagent (Thermo, 15596018). Organoids in Matrigel culture were mechanically disrupted and spun at 400×*g* and 4°C for 5 min. Pellets were re-suspended with PBS, then dissolved in TRIzol. RNA was isolated using miRNeasy kit (QIAGEN, 217004) with on-column DNase digestion (QIAGEN, 79254) according to the manufacturer's instructions.

### Quantitative real-time PCR

The SMARTer cDNA synthesis kit (Clontech, 634925) was used for cDNA synthesis from total RNA following the manufacturer's protocol. Quantitative PCR was performed using the iTag universal SYBR Green supermix (Bio-Rad, 1725121) on the CFX96 Touch real-time PCR detection system. The list of PCR primers is presented in *Supplementary file 1*.

### RNA-seq analysis

RNA was purified from colon cancer organoids using QIAGEN miRNeasy kit (217004) with on-column DNase digestion according to the manufacturer's instructions. RNA-seq libraries were constructed following Illumina TruSeq Stranded mRNA protocol (20020594). The RNA-seq libraries were sequenced

on the Illumina NextSeq 500 (42 bp paired-end run) and Illumina HiSeq 2500 instrument platform (76 bp single-end sequencing) for organoids and colitis-induced cancer cells, respectively.

The sequences were basecalled by Illumina RTA embedded in NextSeq Control Software by the standard workflow. The sequencing reads were aligned onto *Mus musculus* GRCm38/mm10 reference genome using the TopHat v2.1.0 software or Kallisto v0.45.0. The mapped reads were transformed into the count matrix with default parameters using the HTSeq v0.8.0 software, then normalized using the DESeq v2 software. Differentially expressed genes were identified using the same software based on a negative binomial generalized linear model. Sleuth was used to analyze statistical significance of Kallisto data.

## ATAC-seq

Eight days cultured organoids were isolated as single cells using TrypLE (Thermo, 12605010) and fluorescent activated cell sorting (FACS). Dead cells were excluded by Zombie Aqua (BioLegend 423102) staining. Genomic DNAs were extracted by DNA Clean & Concentrator (Zymo). ATAC-seq libraries were constructed with 50K cells from each condition following Omni-ATAC protocol (Illumina 20034197, *Corces et al., 2017*).

Sequenced reads were trimmed with adaptor sequences (cutadapt v1.9.1, *Martin, 2011*) and mapped to the mouse genome (GRCm38, emsembl release 93) by Bowtie2 (v2.3.4.1, *Langmead and Salzberg, 2012*). Mitochondrial and duplicated reads were removed by SAMtools (v1.9, *Li et al., 2009*) and Picard (v2.9.0, https://broadinstitute.github.io/picard/), respectively. Peaks were found by MACS2 (v2.1.1, *Zhang et al., 2008*) and visualized by deepTools (v3.1.1, *Ramírez et al., 2014*, NAR).

## ChIP-seq

Eight days cultured organoids were isolated as single cells using TrypLE (Thermo, 12605010) and washed with PBS three times. Chromatin immunoprecipitation was performed using the iDeal ChIP-seq kit for Transcription Factors (Diagenode, C01010054) and recombinant Anti-HNF4-alpha antibody, ChIP Grade (Abcam, ab181604) following the manufacturer's protocols. ChIP-seq libraries were constructed by Yale Center for Genome Analysis (YCGA) using the KAPA HyperPrep kit and sequenced on the Illumina HiSeq 2500 (150 bp paired-end run).

Sequenced reads were trimmed with adaptor sequences (cutadapt v1.9.1) and mapped to the mouse genome (GRCm38, emsembl release 93) by Bowtie2 (v2.3.4.1). Mitochondrial and duplicated reads were removed by SAMtools (v1.9) and Picard (v2.9.0), respectively. Peaks were found by MACS3 (v3.0.0a6) and visualized by deepTools (v3.1.1).

## Confocal microscopy

Organoids were grown in the Nunc Lab-Tek II Chamber Slide System (Thermo Scientific, 154534PK). Organoids were fixed with freshly prepared 4% PFA in the PME buffer (50 mM PIPES, 2.5 mM $MgCl_2$, 5 mM of EDTA) for 20 min at room temperature. Fixed organoids were permeabilized with PBS containing 0.5% Triton X-100. Fixed and permeabilized organoids were washed with PBS containing 0.05% Tween 20 then blocking was performed using the wash buffer containing 1% BSA. Organoids were stained with anti-LYZ antibody (Clone: Poly28600, BioLegend, 8600001), Alexa Fluor 488-conjugated goat anti-rabbit-IgG polyclonal antibody (Invitrogen, A32731) and DAPI (Sigma, D9542). Pictures were captured using a Zeiss Axio Observer Z.1 microscope or a Zeiss LSM780 confocal laser-scanning microscope.

## Mouse single-cell RNA data processing

scRNA-seq library was generated using Chromium Single Cell 3' Reagent Kits v3 (10x Genomics) following the manufacturer's protocol. Biological replicates were used to generate the replicate library with another methodology as described previously. Fastq files were mapped to pre-built mouse reference set (mm10) and were converted to counts using the pipeline 'cellranger count'. Cells exhibiting percent.mt>30%, nFeatures<200, and nFeatures>8000 were filtered out. The count matrix was log-normalized with a pseudo-count of 1 ('NormalizeData'). The features were then scaled and centered ('ScaleData'). PCA was performed on scaled matrix of 2000 highly variable genes (HVG, 'FindVariableFeatures') with the top 20 principal components (PCs) detected by knee-point. Clustering was

performed by first constructing shared nearest-neighbor graph (SNN) and then applying Louvain with resolution of 0.5 ('FindNeighbors' and 'FindClusters').

## Mouse cell-type annotation

Clusters were annotated based on the expression of canonical marker genes, including *Ptprc* for immune cells, *Epcam*, *Krt7*, and *Klf5* for tumor cells, *Alb*, *Ttr*, and *Crp* for hepatocytes, *Pecam1*, *Eng*, and *Cd34* for endothelial cells.

Tumor cells were than subsetted, subjected to scaling, PCA on 2000 HVG scaled data with the top 20 PCs, and clustering with resolution of 0.4. Among six clusters, cluster 6 exhibited high expression of *Lyz1*, *Muc2*, and *Atoh1* which indicated its Paneth cell and Goblet cell properties.

For cancer stem cell niche analysis, we selected tumor cells with *Lgr5* expression>0. The subsetted dataset were subjected to scaling, PCA on 2000 HVG scaled data with the top 20 PCs, and clustering with resolution of 0.18.

## Human single-cell data processing

We reanalyzed colorectal cancer patient single-cell data (*Joanito et al., 2022*) and utilized cells pre-annotated as epithelial cells. Cells exhibiting percent.mt>25% and nFeatures<200 were filtered out. Additionally, we removed cells with PTPRC expression to make sure our epithelial cells lack immune cells, yielding 47,682 cells. The count matrix was log-normalized with a pseudo-count of 1 ('NormalizeData'). The features were then scaled and centered ('ScaleData'). PCA was performed on scaled matrix of 2000 HVG ('FindVariableFeatures') with the top 20 PCs detected by knee-point. Clustering was performed by first constructing SNN and then applying Louvain with resolution of 0.5 ('FindNeighbors' and 'FindClusters').

## Human regulon analysis

To confirm reduced activity of HNF4A and elevated activity of SOX9 in LYZ+ human colon cancer cells compared to the LYZ- cancer cells, we utilized single-cell regulatory network inference (*Aibar et al., 2017*) and obtained a regulon-activity matrix for each cell. Wilcoxon signed-rank test with p-values adjusted through Bonferroni correction was performed to compare the activity of HNF4A and SOX9 regulon. Further the regulon-activity matrix was z-scaled to generate the heatmap in *Figure 7B*.

## Gene set enrichment analysis

To characterize LYZ+ human colon cancer cells, GSEA used the fGSEA package on HALLMARK canonical pathways in MSigDB v7.1 (*Subramanian et al., 2005*). GSEA was performed on pre-ranked genes using 10,000 permutations. Gene ranks were calculated based on MAST differential expression result applying the following formula:

$$stat = avg_{log}2FCl \left| \left( \frac{avg_{log}2FC}{qnorm\left(p_{val}\right)} \right) \right|$$

For genes with an infinite value of statistics due to a p-value equal to 0 or 1, the statistics was replaced with the highest or lowest value of the statistical metric.

## Paneth cell module score calculation

We computed Paneth cell module score for each cell with marker genes in PanglaoDB (*Franzén et al., 2019*) by implementing 'AddModuleScore' function in Seurat (*Hao et al., 2021*). For human dataset, we filtered out marker genes used in mouse, and for mouse dataset, we filtered out marker genes for human.

# Acknowledgements

The authors would like to thank Gouzel Tokmulina for the FACS.

# Additional information

### Competing interests
Jungmin Choi: Reviewing editor, *eLife*. The other authors declare that no competing interests exist.

### Funding

| Funder | Grant reference number | Author |
| --- | --- | --- |
| National Cancer Institute | RO1 CA168670-01 | Alfred LM Bothwell |
| Yonsei University | Future-Leading Research Initiative 2023-22-0438 | Jae Hun Shin |
| National Research Foundation of Korea | ) RS-2023-00213586 | Jae Hun Shin |

The funders had no role in study design, data collection and interpretation, or the decision to submit the work for publication.

### Author contributions
Jae Hun Shin, Conceptualization, Resources, Data curation, Software, Formal analysis, Supervision, Funding acquisition, Validation, Investigation, Visualization, Methodology, Writing – original draft, Project administration, Writing – review and editing; Jooyoung Park, Jaechul Lim, Resources, Data curation, Software, Formal analysis, Validation, Investigation, Visualization, Methodology, Writing – original draft, Writing – review and editing; Jaekwang Jeong, Resources, Formal analysis, Investigation, Methodology, Writing – original draft; Ravi K Dinesh, John Wysolmerski, Resources; Stephen E Maher, Jun Young Hong, Resources, Investigation; Jeonghyun Kim, Soyeon Park, Investigation; Jungmin Choi, Resources, Software, Validation, Visualization, Methodology, Writing – original draft, Writing – review and editing; Alfred LM Bothwell, Conceptualization, Supervision, Validation, Writing – original draft, Writing – review and editing

### Author ORCIDs
Jae Hun Shin https://orcid.org/0000-0001-6066-0017
Jooyoung Park https://orcid.org/0000-0001-9001-6388
Jaechul Lim https://orcid.org/0000-0002-6075-2656
Jungmin Choi https://orcid.org/0000-0002-8614-0973

### Ethics
All animal studies were conducted with the approval of the Institutional Animal Care and Use Committee at the Yale University School of Medicine (protocol #2020-11211). In this research, 8-12 week old C57BL/6 mice were utilized for colon cancer organoid transplantation via splenic injection, to develop liver metastasis. All procedures were performed in accordance with ethical guidelines to ensure the welfare and humane treatment of the animals involved in the study.

Reviewer #1 (Public review): https://doi.org/10.7554/eLife.97279.3.sa1
Reviewer #2 (Public review): https://doi.org/10.7554/eLife.97279.3.sa2
Author response https://doi.org/10.7554/eLife.97279.3.sa3

# Additional files

### Supplementary files
• MDAR checklist
• Supplementary file 1. The list of primers used in quantitative real time PCR.

### Data availability
All data are included in the article and *Supplementary file 1*. All the sequencing data reported in this study have been deposited in the Gene Expression Omnibus as follows; Bulk RNA-seq: GSE157531, ATAC-seq: GSE157529, Chip-seq: GSE277510, scRNA-seq: GSE157645.

The following datasets were generated:

| Author(s) | Year | Dataset title | Dataset URL | Database and Identifier |
|---|---|---|---|---|
| Pekkarinen M, Nordfors K , Uusi-Mäkelä J, Kytölä V, Hartewig A, Huhtala L, Rauhala M, Urhonen H, Häyrynen S, Afyounian E, Yli-Harja O, Zhang W, Helen P, Lohi O, Haapasalo H, Haapasalo J, Nykter M, Kesseli J, Rautajoki KJ | 2024 | Aberrant DNA methylation distorts developmental trajectories in atypical teratoid/rhabdoid tumors | https://www.ncbi.nlm.nih.gov/geo/query/acc.cgi?acc=GSE197569 | NCBI Gene Expression Omnibus, GSE197569 |
| Shin JH, Choi J, Bothwell A | 2021 | Dickkopf-2 regulates stemness and differentiation of colorectal cancer cells via c-Src and HNF4α1 | https://www.ncbi.nlm.nih.gov/geo/query/acc.cgi?acc=GSE157645 | NCBI Gene Expression Omnibus, GSE157645 |
| Shin JH | 2024 | Metastasis of colon cancer requires Dickkopf-2 to generate cancer cells with Paneth cell properties | https://www.ncbi.nlm.nih.gov/geo/query/acc.cgi?acc=GSE277510 | NCBI Gene Expression Omnibus, GSE277510 |
| Shin JH | 2024 | Metastasis of colon cancer requires Dickkopf-2 to generate cancer cells with Paneth cell properties | https://www.ncbi.nlm.nih.gov/geo/query/acc.cgi?acc=GSE157529 | NCBI Gene Expression Omnibus, GSE157529 |

The following previously published datasets were used:

| Author(s) | Year | Dataset title | Dataset URL | Database and Identifier |
|---|---|---|---|---|
| Joanito I, Wirapati P, Zhao N, Nawaz Z, Yeo G, Lee F, Eng CLP, Macalinao DC, Kahraman M, Srinivasan H, Lakshmanan V, Verbandt S, Tsantoulis P, Gunn N, Venkatesh PN, Poh ZW, Nahar R, Oh HLJ, Loo JM, Chia S, Cheow LF, Cheruba E, Wong MT, Kua L, Chua C, Nguyen A, Golovan J, Gan A, Lim WJ, Guo YA, Yap CK, Tay B, Hong Y, Chong DQ, Chok AY, Park WY, Han S, Chang MH, Seow-En I, Fu C, Mathew R, Toh EL, Hong LZ, Skanderup AJ, DasGupta R, Ong CJ, Lim KH, Tan EKW, Koo SL, Leow WQ, Tejpar S, Prabhakar S, Tan IB | 2022 | Single cell RNA sequencing of colorectal cancer patients (CRC-SG1) | https://ega-archive.org/datasets/EGAD00001008555 | European Genome-Phenome Archive, EGAD00001008555 |

*Continued on next page*

*Continued*

| Author(s) | Year | Dataset title | Dataset URL | Database and Identifier |
|---|---|---|---|---|
| Joanito I, Wirapati P, Zhao N, Nawaz Z, Yeo G, Lee F, Eng CLP, Macalinao DC, Kahraman M, Srinivasan H, Lakshmanan V, Verbandt S, Tsantoulis P, Gunn N, Venkatesh PN, Poh ZW, Nahar R, Oh HLJ, Loo JM, Chia S, Cheow LF, Cheruba E, Wong MT, Kua L, Chua C, Nguyen A, Golovan J, Gan A, Lim WJ, Guo YA, Yap CK, Tay B, Hong Y, Chong DQ, Chok AY, Park WY, Han S, Chang MH, Seow-En I, Fu C, Mathew R, Toh EL, Hong LZ, Skanderup AJ, DasGupta R, Ong CJ, Lim KH, Tan EKW, Koo SL, Leow WQ, Tejpar S, Prabhakar S, Tan IB | 2022 | Single cell RNA sequencing of colorectal cancer patients (KUL3) | https://ega-archive. org/datasets/ EGAD00001008584 | European Genome-Phenome Archive, EGAD00001008584 |
| Joanito I, Wirapati P, Zhao N, Nawaz Z, Yeo G, Lee F, Eng CLP, Macalinao DC, Kahraman M, Srinivasan H, Lakshmanan V, Verbandt S, Tsantoulis P, Gunn N, Venkatesh PN, Poh ZW, Nahar R, Oh HLJ, Loo JM, Chia S, Cheow LF, Cheruba E, Wong MT, Kua L, Chua C, Nguyen A, Golovan J, Gan A, Lim WJ, Guo YA, Yap CK, Tay B, Hong Y, Chong DQ, Chok AY, Park WY, Han S, Chang MH, Seow-En I, Fu C, Mathew R, Toh EL, Hong LZ, Skanderup AJ, DasGupta R, Ong CJ, Lim KH, Tan EKW, Koo SL, Leow WQ, Tejpar S, Prabhakar S, Tan IB | 2022 | Single cell RNA sequencing of colorectal cancer patients (KUL5) | https://ega-archive. org/datasets/ EGAD00001008585 | European Genome-Phenome Archive, EGAD00001008585 |

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
