## [Editor Report · eLife Assessment]

This **valuable** study proposes that protein secreted by colon cancer cells induces cells with Paneth-like properties that favor colon cancer metastasis. The evidence supporting the conclusions is **solid** but the study would benefit from more direct experiments to test the functional role of Paneth-like cells and to monitor metastasis from colon tumors. The work will be of interest to researchers studying colon cancer metastasis.

---

## [Referee Report · Reviewer #1 (Public review)]

Summary:

The authors addressed the influence of DKK2 on colorectal cancer (CRC) metastasis to the liver using an orthotopic model transferring AKP-mutant organoids into the spleens of wild-type animals. They found that DKK2 expression in tumor cells led to enhanced liver metastasis and poor survival in mice. Mechanistically, they associate Dkk2-deficiency in donor AKP tumor organoids with reduced Paneth-like cell properties, particularly Lz1 and Lyz2, and defects in glycolysis. Quantitative gene expression analysis showed no significant changes in Hnf4a1 expression upon Dkk2 deletion. Ingenuity Pathway Analysis of RNA-Seq data and ATAC-seq data point to a Hnf4a1 motif as a potential target. They also show that HNF4a binds to the promoter region of Sox9, which leads to LYZ expression and upregulation of Paneth-like properties. By analyzing available scRNA data from human CRC data, the authors found higher expression of LYZ in metastatic and primary tumor samples compared to normal colonic tissue; reinforcing their proposed link, HNF4a was highly expressed in LYZ+ cancer cells compared to LYZ- cancer cells.

Strengths:

Overall, this study contributes a novel mechanistic pathway that may be related to metastatic progression in CRC.

Weaknesses:

The main concerns are related to incremental gains, missing in vivo support for several of their conclusions in murine models, and missing human data analyses.

Main comments

Novelty:

The authors previously described the role of DKK2 in primary CRC, correlating increased DKK2 levels to higher Src phosphorylation and HNF4a1 degradation, which in turn enhances LGR5 expression and "stemness" of cancer cells, resulting in tumor progression (PMID: 33997693). A role for DKK2 in metastasis has also been previously described (sarcoma, PMID: 23204234)

Mouse data:

(a) The authors analyzed liver mets, but the main differences between AKT and AKP/Dkk2 KO organoids could arise during the initial tumor cell egress from the intestinal tissue (which cannot be addressed in their splenic injection model), or during pre-liver stages, such as endothelial attachment. While the analysis of liver mets is interesting, given that Paneth cells play a role in the intestinal stem cell niche, it is questionable whether a study that does not involve the intestine can appropriately address this pathway in CRC metastasis.

(b) The overall number of Paneth cells found in the scRNA-seq analysis of liver mets was low (17 cells, Fig.3), and assuming that these cells are driving the differences seems somewhat far-fetched.

(c) Fig. 6 suggests a signaling cascade in which the absence of DKK2 leads to enhanced HNF4A expression, which in turn results in reduced Sox9 expression and hence reduced expression of Paneth cell properties. It is therefore crucial that the authors perform in vivo (splenic organoid injection) loss-of-function experiments, knockdown of Sox9 expression in AKP organoids, and Sox9 overexpression experiments in AKP/Dkk2 KO organoids to demonstrate Sox9 as the central downstream transcription factor regulating liver CRC metastasis.

(d) Given the previous description of the role of DKK2 in primary CRC, it is important to define the step of liver metastasis affected by Dkk2 deficiency in the metastasis model. Does it affect extravasation, liver survival, etc.?

Human data:

Can the authors address whether the expression of Dkk2 changes in human CRC and whether mutations in Dkk2 as correlated with metastatic disease or CRC stage?

Bioinformatic analysis

GEO repositories remain not open (at the time of the re-review) and SRA links for raw data are still unavailable. Without access to raw data, it is not possible to verify the analyses or fully assess the results. A part of the article was made by re-analyzing public data so the authors should make even the raw available and not just the count tables

---

## [Referee Report · Reviewer #2 (Public review)]

Summary:

The authors propose that DKK2 is necessary for the metastasis of colon cancer organoids. They then claim that DKK2 mediates this effect by permitting the generation of lysozyme-positive Paneth-like cells within the tumor microenvironmental niche. They argue that these lysozyme-positive cells have Paneth-like properties in both mouse and human contexts. They then implicate HNF4A as the causal factor responsive to DKK2 to generate lysozyme-positive cells through Sox9.

Strengths:

The use of a genetically defined organoid line is state-of-the-art. The data in Figure 1 and the dependence of DKK2 for splenic injection and liver engraftment, as well as the long-term effect on animal survival, are interesting and convincing. The rescue using DKK2 administration for some of their phenotype in vitro is good. The inclusion and analysis of human data sets help explore the role of DKK2 in human cancer and help ground the overall work in a clinical context.

Remaining Weaknesses after revision:

(1) The authors have effectively explained the regulation of HNF4A at both mRNA and protein levels. To further strengthen their findings, I recommend using CRISPR technology to generate DKK2 and HNF4A double knockout organoids. This approach would allow the authors to investigate whether the AKP liver metastasis is restored in the double knockout condition. Such an experiment would provide more direct evidence that HNF4A protein stabilization is the crucial mechanism for liver metastasis suppression following DKK2 knockout.

---

## [Author Response]

The following is the authors’ response to the original reviews.

**Public Reviews:**

**Reviewer #1 (Public Review):**
Summary:The authors addressed the influence of DKK2 on colorectal cancer (CRC) metastasis to the liver using an orthotopic model transferring AKP-mutant organoids into the spleens of wild-type animals. They found that DKK2 expression in tumor cells led to enhanced liver metastasis and poor survival in mice. Mechanistically, they associate Dkk2-deficiency in donor AKP tumor organoids with reduced Paneth-like cell properties, particularly Lz1 and Lyz2, and defects in glycolysis. Quantitative gene expression analysis showed no significant changes in Hnf4a1 expression upon Dkk2 deletion. Ingenuity Pathway Analysis of RNA-Seq data and ATAC-seq data point to a Hnf4a1 motif as a potential target. They also show that HNF4a binds to the promoter region of Sox9, which leads to LYZ expression and upregulation of Paneth-like properties. By analyzing available scRNA data from human CRC data, the authors found higher expression of LYZ in metastatic and primary tumor samples compared to normal colonic tissue; reinforcing their proposed link, HNF4a was highly expressed in LYZ+ cancer cells compared to LYZ- cancer cells.Strengths:Overall, this study contributes a novel mechanistic pathway that may be related to metastatic progression in CRC.Weaknesses:The main concerns are related to incremental gains, missing in vivo support for several of their conclusions in murine models, and missing human data analyses. Additionally, methods and statistical analyses require further clarification.Main comments:(1) NoveltyThe authors previously described the role of DKK2 in primary CRC, correlating increased DKK2 levels to higher Src phosphorylation and HNF4a1 degradation, which in turn enhances LGR5 expression and "stemness" of cancer cells, resulting in tumor progression (PMID: 33997693). A role for DKK2 in metastasis has also been previously described (sarcoma, PMID: 23204234).(2) Mouse dataa) The authors analyzed liver mets, but the main differences between AKT and AKP/Dkk2 KO organoids could arise during the initial tumor cell egress from the intestinal tissue (which cannot be addressed in their splenic injection model), or during pre-liver stages, such as endothelial attachment. While the analysis of liver mets is interesting, given that Paneths cells play a role in the intestinal stem cell niche, it is questionable whether a study that does not involve the intestine can appropriately address this pathway in CRC metastasis.

We value the reviewer’s comment that the splenic injection model cannot represent metastasis from the primary tumors, intravasation and extravasation. Therefore, we performed the orthotopic transplantation of AKP and KO organoids into the colon directly then, tested metastasis of cancer.

**Author response image 1. sa3fig1:** Primary tumor formation and liver metastasis by orthotopic transplantation of AKP or KO colon cancer organoids. 6-8 week-old male C57BL/6J mice were treated with 2.5% DSS dissolved in drinking water for 5 days, followed by regular water for 2 days to remove gut epithelium. After recovery with the regular water, the colon was flushed with 1000 μl of 0.1% BSA in PBS. Then, 200,000 dissociated organoid cells in 200 μl of 5% Matrigel and 0.1% BSA in PBS were instilled into the colonic luminal space. After infusion, the anal verge was sealed with Vaseline. 8 weeks after transplantation, the mice were sacrificed to measure primary tumor formation and liver metastasis.

As a result, 4 out 6 mice in the control group successfully formed colorectal primary tumors whereas only 2 out 6 mice showed primary tumor formation in the KO group (Author response image 1A). The size of tumors was reduced by about half (10-12 mm to 5-7 mm). Only one AKP mouse developed metastasized nodules in the liver (Author response image 1B). Next, to measure the circulating tumor cells, we harvested at least 500 ul of bloods from the portal vein and then analyzed tdTomato-positive tumor cells (Author response image 2). Flow cytometry analysis of PBMCs showed the presence of tdTomatohiCD45- cells as well as tdTomatomidCD45+ cells in 2 out of 6 AKP mice, while no tdTomato-positive cells were observed in the PBMCs of KO organoid-transplanted mice.

Due to the limited numbers of mice showed primary and metastatic tumor formation, we cannot provide a statistic analysis of DKK2-mediated metastasis. However, our revised data indicate a trend that DKK2 KO reduced primary tumor formation, the number of circulating tumor cells and liver metastasis. This trend is consistent with our previous report in the *iScience* paper, which showed that DKK2 KO reduced AOM/DSS-induced polyp formation about 60 % and decreased metastasis in the splenic injection model system in this manuscript. Further studies are necessary to confirm this trend and to provide the underlying mechanisms of intravasation and extravasation of circulating tumor cells.

**Author response image 2. sa3fig2:** Flow cytometry analysis of tdTomato+ circulating colon tumor cells in PBMCs. PBMCs were harvested via the portal vein after euthanasia. CD45 and tdTomato were analyzed by flow cytometry.

b) The overall number of Paneth cells found in the scRNA-seq analysis of liver mets was strikingly low (17 cells, Figure 3), and assuming that these cells are driving the differences seems somewhat far-fetched. Adding to this concern is inappropriate gating in the flow plot shown in Figure 6. This should be addressed experimentally and in the interpretation of data.

We appreciate for reviewer’s comments to clarify this point. Since the number of LYZ+ cells is low in our scRNA-seq analysis, we performed flow cytometry in Figure 6H showing the clear population expressing LYZ in the same splenic injection model of metastasis. Figure 6H is a representative image of triplicates for each group and we performed this experiment three times, independently. As suggested, we changed the graph format and updated the gating and statistical analysis in Fig 6H and 6I. This in vivo result confirmed our in vitro data showing that DKK2 KO reduced LYZ+ cells while increase the HNF4α1 proteins.

c) Figures 3, 5, and 6 show the individual gene analyses with unclear statistical data. It seems that the p-values were not adjusted, and it is unclear how they reached significance in several graphs. Additionally, it was not stated how many animals per group and cells per animal/group were included in the analyses.

In Fig. 3, mouse scRNA-seq data were generated from pooled cancer samples from 5 animals per group. The Wilcoxon signed-rank test was performed for each gene and/or regulon activity. Since multiple testing adjustments were not performed, a p-value adjustment is neither needed nor applicable..

In Fig. 5, human data were analyzed. Cells from the same sample are dependent, but differential gene expression (DEG) analysis typically calculates statistics under the assumption that they are independent. This assumption may explain the low p-values observed in our data. To address this issue, we applied pseudobulk DEG analysis to our human single-cell data. Even after correcting for statistical error, we confirmed that the genes of interest still exhibited significantly different expression patterns (Author response image 3).

**Author response image 3. sa3fig3:** Pseudobulk DEG analysis confirmed the differential expression genes of interest.

In Fig.6H-6I, the number of animals per group is provided in the figure legend.

d) Figure 6 suggests a signaling cascade in which the absence of DKK2 leads to enhanced HNF4A expression, which in turn results in reduced Sox9 expression and hence reduced expression of Paneth cell properties. It is therefore crucial that the authors perform in vivo (splenic organoid injection) loss-of-function experiments, knockdown of Sox9 expression in AKP organoids, and Sox9 overexpression experiments in AKP/Dkk2 KO organoids to demonstrate Sox9 as the central downstream transcription factor regulating liver CRC metastasis.

Sox9 is a well-established marker gene for Paneth cell formation in the gut. Therefore, overexpression or knockout of the Sox9 gene would result in either an increase or decrease in Paneth cells in the organoids. We believe that the suggested experiments fall outside the scope of this manuscript. Instead, we demonstrated the change in the Paneth cell differentiation marker, Sox9, in the presence or absence of DKK2.

e) Given the previous description of the role of DKK2 in primary CRC, it is important to define the step of liver metastasis affected by Dkk2 deficiency in the metastasis model. Does it affect extravasation, liver survival, etc.?

We appreciate the reviewer’s insights and perspectives. Regarding liver survival, it is well known that stem cell niche formation is a critical step for the outgrowth of metastasized cancer cells (Fumagalli et al. 2019, Cell Stem Cell). LYZ+ Paneth cells are recognized as stem cell niche cells in the intestine, and human scRNA-seq data have shown that LYZ+ cancer cells express stem cell niche factors such as Wnt and Notch ligands. To determine whether LYZ+ cancer cells act as stem cell niche cells, we performed confocal microscopy to assess whether LYZ+ cancer cells express WNT3A and DLL4 in AKP organoids (Author response image 4). The results show that LYZ labeling co-localizes with DLL4 and WNT3A expression, while the organoid reporter tdTomato is evenly distributed. Additionally, our in vitro and in vivo data indicate that DKK2 deficiency leads to a reduction of LYZ+ cancer cells, which may contribute to stem cell niche formation. Based on these findings, we propose that DKK2 is an essential factor for stem cell niche formation, which is required for cancer cell survival in the liver during the early stages of metastasis. Although our revised data confirmed the trend that DKK2 deficiency decreases liver metastasis, we have not yet determined whether DKK2 is involved in extravasation. This research topic should be addressed in future studies.

**Author response image 4. sa3fig4:** Confocal microscopy analysis for lysozyme (LYZ) and Paneth cell-derived stem cell niche factors, WNT3A and DLL4 in AKP colon cancer organoids.

The method is described in the supplemental information. The list of antibodies used: DLL4 (delta-like 4) Polyclonal Antibody (Invitrogen, PA5-85931), WNT3A Polyclonal Antibody (Invitrogen, PA5-102317), Goat anti-Rabbit IgG (H+L) Cross-Adsorbed Secondary Antibody, Alexa Fluor 488 (Invitrogen, A-11008), Anti-Lysozyme C antibody (H-10, Santacurz, sc-518083), Goat anti-Mouse IgM (Heavy chain) Secondary Antibody, Alexa Fluor 647 (Invitrogen, A-21238).

(3) Human dataCan the authors address whether the expression of Dkk2 changes in human CRC and whether mutations in Dkk2 as correlated with metastatic disease or CRC stage?

The human data were useful in identifying the presence of LYZ+ cancer cells with Paneth cell properties. However, due to the limited number of late-stage patient samples with high DKK2 expression, the results were not statistically significant. Nevertheless, the trend suggests a positive correlation between DKK2 expression and the malignant stage of CRC.

(4) Bioinformatic analysisThe authors did not provide sufficient information on bioinformatic analyses. The authors did not include information about the software, cutoffs, or scripts used to make their analyses or output those figures in the manuscript, which challenges the interpretation and assessment of the results. Terms like "Quantitative gene expression analyses" (line 136) "visualized in a Uniform Approximation and Projection" (line 178) do not explain what was inputted and the analyses that were executed. There are multiple forms to align, preprocess, and visualize bulk, single cell, ATAC, and ChIP-seq data, and depending on which was used, the results vary greatly. For example, in the single-cell data, the authors did not inform how many cells were sequenced, nor how many cells had after alignment and quality filtering (RNA count, mt count, etc.), so the result on Paneth+ to Goblet+ percent in lines 184 and 185 cannot be reached because it depends on this information. The absence of a clustering cutoff for the single-cell data is concerning since this greatly affects the resulting cluster number (https://www.nature.com/articles/s41592-023-01933-9). The authors should provide a comprehensive explanation of all the data analyses and the steps used to obtain those results.

We apologize for the insufficient information. Below, we provide detailed information on the data analyses, which are also available in the GEO database (Bulk RNA-seq: GSE157531, ATAC-seq: GSE157529, ChIP-seq: GSE277510). Methods are updated in the current version of supplemental information.

(5) Clarity of methods and experimental approachesThe methods were incomplete and they require clarification.

We’ve updated our methods as requested by the reviewer.

**Reviewer #2 (Public Review):**
Summary:The authors propose that DKK2 is necessary for the metastasis of colon cancer organoids. They then claim that DKK2 mediates this effect by permitting the generation of lysozyme-positive Paneth-like cells within the tumor microenvironmental niche. They argue that these lysozyme-positive cells have Paneth-like properties in both mouse and human contexts. They then implicate HNF4A as the causal factor responsive to DKK2 to generate lysozyme-positive cells through Sox9.Strengths:The use of a genetically defined organoid line is state-of-the-art. The data in Figure 1 and the dependence of DKK2 for splenic injection and liver engraftment, as well as the long-term effect on animal survival, are interesting and convincing. The rescue using DKK2 administration for some of their phenotype in vitro is good. The inclusion and analysis of human data sets help explore the role of DKK2 in human cancer and help ground the overall work in a clinical context.Weaknesses:In this work by Shin et al., the authors expand upon prior work regarding the role of Dickkopf-2 in colorectal cancer (CRC) progression and the necessity of a Paneth-like population in driving CRC metastasis. The general topic of metastatic requirements for colon cancer is of general interest. However, much of the work focuses on characterizing cell populations in a mouse model of hepatic outgrowth via splenic transplantation. In particular, the concept of Paneth-like cells is primarily based on transcriptional programs seen in single-cell RNA sequencing data and needs more validation. Although including human samples is important for potential generality, the strength could be improved by doing immunohistochemistry in primary and metastatic lesions for Lyz+ cancer cells. Experiments that further bolster the causal role of Paneth-like CRC cells in metastasis are needed.
**Recommendations for the Authors:**

**Reviewing Editor (Recommendations for the Authors):**
Here we note several key concerns with regard to the main conclusions of the paper. Additional experiments to directly address these concerns would be required to substantially update the reviewer evaluation.(1) Demonstration of a causal role of Paneth-like cells in CRC metastasis, for example by sorting the Paneth-like cells - either by the markers they identified in the subsequent single cell or by scatter - to establish whether the frequency of the Paneth-like cells in a culture of organoids is directly correlated with tumorigenicity and engraftment.

We sincerely appreciate the reviewing editor’s comment. First, as previously reported (Shin et al., iScience 2021), there is no difference in proliferation between WT and KO during in vitro organoid culture or in vivo colitis-induced tumors. However, DKK2 deficiency led to morphological changes, which we analyzed using bulk RNA-seq. As described in the manuscript, Paneth cell marker genes, such as Lysozymes and defensins, were significantly reduced in DKK2 KO AKP organoids.

Due to the nature of these markers, it is technically challenging to isolate live LYZ+ cancer cells. To address this issue in the future, we plan to develop organoids that express a reporter gene specific for Paneth cells. In this manuscript, we demonstrated a correlation between DKK2 and the formation of LYZ+ cancer cells. In both the splenic injection model (Fig. 1) and the orthotopic transplantation model (Fig. R1-R2), we observed that transplantation of cancer organoids with reduced numbers of LYZ+ cells (KO organoids) led to decreased metastatic tumor formation. The number of LYZ+ cells in KO-transplanted mice remained low in liver metastasized tumor nodules (Fig. 6H-I6). Immunohistochemistry further confirmed that LYZ+ cancer cells were barely detectable in KO samples (Author response image 5). These data suggest that DKK2 is essential for the formation of LYZ+ cancer cells, which are necessary for outgrowth following metastasis.

**Author response image 5. sa3fig5:** Histology of Lysozyme positive cells in metastasized tumor nodules in liver of colon cancer organoid transplanted mice. Immunohistochemistry of Lysozyme positive Paneth-like cells cells in liver metastasized colon cancer (Upper panels, DAB staining). Identification of tumor nodules by H&E staining (lower panels, Scale bar = 100 μm). Magnified tumor nodules are shown in the 2nd and 3rd columns (Scale bar = 25 μm). Arrows indicate Lysozyme positive Paneth like cells in tumor epithelial cells. Infiltration of Lysozyme positive myeloid cells is detected in both AKP and KO tumor nodules. AKP: Control colon cancer organoids carrying mutations in Apc, Kras and Tp53 genes. KO: Dkk2 knockout colon cancer organoids

(2) Further characterization of Lyz+/Paneth-like cells to further the authors' argument for the unique function that they have in their tumor model. Specifically, do the cells with Paneth-like cells secrete Wnt3, EGF, Notch ligand, and DII4 as normal Paneth cells do?

We appreciate the reviewing editor’s comment. In response, we performed confocal microscopy analysis to examine the protein levels of LYZ, Wnt3A, and DLL4 in AKP colon cancer organoids (Author response image 4). The data presented above show that LYZ+ cancer cells express both Wnt3A and DLL4, suggesting that LYZ+ colon cancer cells may function similarly to Paneth cells, which are stem cell niche cells. Furthermore, using the Panglao database, we demonstrated that LYZ+/Paneth-like cells exhibit typical Paneth cell properties in human scRNA-seq data (Fig. 4 and Fig. 5). These findings suggest that LYZ+ colon cancer cells possess Paneth cell properties.

(3) Experiments to test metastasis, ideally from orthotopic colonic tumors, to ensure phenotypes aren't restricted to the splenic model of hepatic colonization and outgrowth used at present.

We are in agreement with the reviewing editor and reviewers, which is why we conducted the orthotopic transplantation experiment. However, we encountered challenges in establishing this model effectively. After multiple trials, we observed that many mice did not form primary tumors, and the variability, particularly in metastasis, was difficult to control. Only a few AKP-transplanted mice developed liver metastasis. The representative revision data have been provided above. Nevertheless, we believe that this model needs further improvement and optimization to reliably study metastasis originating from primary tumors.

(4) To generalize claims to human cancer, the authors should test whether loss of DKK2 impacts LYZ+ cancer cells in human organoids and affects their engraftment in immunodeficient mice compared to control. Another more correlative way to validate the LYZ+ expression in human colon cancer would be to stain for LYZ in metastatic vs. primary colon cancer, expecting metastatic lesions to be enriched for LYZ+ cells.

We agree with your point, and this will be addressed in future studies.

(5) Clarifying inconsistencies regarding effect of DKK2 loss on HNF4A (Figure 1E vs Figure 6I).

In Figure 1E, we measured the mRNA levels of HNF4A in metastasized foci by qPCR while in Figure 6I, we measured the protein level of HNF4A by flow cytometry. Recent studies, including our previous report, have shown that HNF4A protein levels are regulated by proteasomal degradation mediated by pSrc (*Mori-Akiyama et al. 2007, Gastroenterology, Bastide et al. 2007, Journal of Cell Biology, Shin et al. 2021 iScience*). Consequently, while the mRNA levels remained unchanged in Fig. 1E, we observed a reduction of HNF4A protein levels in Figure 6I.

(6) Addressing concerns about statistics and reporting as outlined by Reviewer 1.

Thank you very much for your assistance in improving our manuscript. The updates have been incorporated as detailed above.

These are the central reviewer concerns that would require additional experimentation to update the editorial summary. Other concerns should be addressed in a revision response but do not require additional experimentation.
**Reviewer #1 (Recommendations For The Authors):**
Specific comments:• Do Dkk2-KO organoids grow normally?

Yes, *in vitro*.

Since the authors reported on the effects of Dkk2 in the induction/maintenance of the Paneth cell niche, changes in AKP organoid numbers of growth rate between Dkk2-WT and KO would be an expected outcome.

Disruption of Paneth cell formation in normal organoids is expected to alter growth. However, DKK2 KO in colon cancer organoids with mutations in the *Apc*, *Kras*, and *Tp53* genes exhibits growth rates and organoid sizes similar to those of WT AKP controls. In contrast to *in vitro* observations, we observed a significant reduction in metastasized tumor growth in vivo. Further analyses of factors derived from LYZ+ cancer cells will help address the discrepancy in DKK2's absence between in vitro and in vivo conditions.

• Figure 1:- Panel C: The legend indicates what c.p. stands for.

c.p.m. stands for count per minutes for in vivo imaging analysis. This has been updated in the Figure legend.

- Panel E: Please comment on the possible underlying reasons for the lack of change in HNF4a1 levels.

This has been updated in response to the reviewing editor’s comment (5) above.

- Panel E: Number of mice from which isolated cancer nodules were harvested.

Total mice per group were 5. This has been updated in the legend.

• Figure 2:- Suggestion: Panel A should be presented in Figure 1 since Dkk2 KO organoids are already used in Figure 1.

We added this to present the recovery of DKK2 by adding recombinant DKK2 proteins in Fig.2.

- Panel B: Please explain why these genes are marked in blue.

It has been described in the legend. “Paneth cell marker genes are highlighted as blue circles (AKP=3 and KO=5 biological replicates were analyzed).”

• Figure 3:- Indicate the number of cells recovered from AKP vs. KO mice (since liver metastasis was already reduced in KO mice). This should be shown in a UMAP.- Panel A: 4th line in the pathways, correct "Singel" typo.

We appreciate your correction. It has been fixed.

- Panel A: There are multiple versions of PanglaoDB with different markers; a list of all that was used to determine cell type should be provided.- Panel C: Bar value for the WNT pathway is not displayed, and there is no legend to indicate the direction of the analysis (that is, AKPvsKO or KOvsAKP).

It is KOvsAKP, described in the figure legend.

- Panel C: Ingenuity pathway analysis is not a good tool to look at this type of result because it does not include the gene fold changes in the analysis, so it only provides a Z-score of the presence of that pathway and not the degree it is increased or fold changes - recommend substituting any type of GSEA analysis, such as fgsea. -o Panel D: the term "Patient" to refer to mice is confusing. Use "Mice" or "Treatment" or "Condition" instead.

Corrected

- Panel D: Information about the number of mice per group, cells per animal (or liver let) used, and additional clarification about the statistical analysis used is required, as differences shown in this panel appear subtle given the standard variation in each group. Box plots need to show individual/raw values.• Figure 4:- Panel E: It would be helpful to show the cutoff lines for the Paneth cell score and Lyz expression in the graphs.

It has been updated in response to the reviewer’s request.

• Figure 5:- Panel B: again, information about the number of "patients" or cells used and clarification about the statistical analysis used is required as the display of data generates concerns about the distribution within groups. Box plots need to show individual/raw values

It has been updated in response to the reviewer’s request.

• Figure 6:- Panel A: Add a legend to inform the direction of the process (e.g., red, activation, blue, repression). We noticed the Yap1 bar data had no color. Is there a reason for that? Please explain this point in the revised manuscript.

Red color added for the Yap1.

- Panel A: Ingenuity pathway analysis is not a good tool to look at this type of results because it does not include the gene Foldchanges in the analysis, so it only provides a Z-score of the presence of that pathway and not the degree it is increased or not. I recommend substituting any type of GSEA analysis, such as fgsea.- Panels A&B: Again, only p-value scores were provided, while fold changes are necessary to define the ratio of presence increase of normal vs. AKP.- Panel D: No raw or pre-processed ChIP-seq data was provided. Additionally, please indicate exactly the genome location it seems the image was edited from a raw made on UCSC genome browser-it should be remade by adding coordinates and other important information (genes around, epigenetic, etc.).- Panel H/I: Flow cytometry gating is inappropriate, as its catching cells are negative for LYZ in both AKP and KO cells, resulting in an overestimation of the number of Lyz cells. Gating should specifically select very few LYZ-positive cells in the top/left quadrant.

The updates have been made, and the statistical data have been re-analyzed.

- Panel J: Information about the number of animals/organoids or cells used and clarification about the statistical analysis used is required, as the display of data generates concerns about the distribution within groups. Box plots need to show individual/raw values.• Overall:- A supplementary table with all the sequenced libraries and their depth, read length/cell count should be provided.

All of the information is now available in the GEO database. We used previously published human epithelial datasets for human single cell analysis (Joanito*, Wirapati*, Zhao*, Nawaz* et al, Nat Genetics, 2022, PMID: 35773407).

- The Hallmark Geneset used is very broad, and the authors should confirm the results on GO bp.

Using Gene Ontology biological processes (GO bp), we observed that glycolysis-related genes were enriched in our newly described cell population, although the adjusted p-value did not exceed 0.05.

**Author response image 6. sa3fig6:** GSEA with GOBP pathway highlighted glycoprotein and protein localization to extracellular region, both of which are related Paneth cell functions. Paneth cells secrete α-defensins, angiogenin-4, lysozyme and secretory phospholipase A2. The enriched glycoprotein process and protein localization not extracellular region reflect the characteristics of Paneth cells.

- qPCR is not a good way to confirm sequencing results; while PCR data is pre-normalized, sequencing is normalized only after quantification, so results on 6 E and F should be shown on the sequencing data.

The expression level of Sox9 is relatively low. In our bulk RNA-seq data, the averages for Sox9 in AKP versus DKK2 KO are 28.2 and 25.1, respectively. While there is a similar trend, the difference is not statistically significant in this dataset, and we did not include an experimental group for reconstitution. Therefore, we conducted qPCR experiments for the reconstitution study by adding recombinant DKK2 (rmDKK2) protein to the culture. Furthermore, it is well established that Sox9 is an essential transcription factor for the formation of LYZ+ Paneth cells. Based on this, we assessed the levels of LYZ and Sox9 using qPCR and confocal microscopy in the presence or absence of DKK2.

• Edits in the text:- There are several typographical errors. Specific suggestions are provided below.- Line 43: "Chromatin immunoprecipitation followed by sequencing analysis," state analysis of what cells before continuing with "revealed..." revealed...- Line 77: Recent findings have identified- Line 138: were reduced in KO tumor samples à rephrase to clarify "KO-derived liver tumors"- Line 167: Recombinant mouse DKK2 protein treatment in KO organoids partially rescued this effect. Add "partially" since adding rmDkk2 didn't fully restore Lyz1 and Lyz2 levels.- Line 185-187: the authors should not reference Figure 6 because it has not been introduced yet.- Line 198-199: The authors claimed a correlation between Dkk2 expression and Lgr5 expression; however, the graph presented in Figure 3B does not indicate this. The R-value was 0.11, which does not indicate a correlative expression between these genes.- Line 232-233: the authors need to show any connection to Dkk2 gene expression in human samples in order to draw that conclusion.- Line 294: expression, leading to the formation- Line 347: Wnt ligand (correct Wng typo)

We have modified our manuscript in accordance with the reviewer’s suggestions.

**Reviewer #2 (Recommendations For The Authors):**
Specific criticisms/suggestions:Author claim 1: Dkk2 is necessary for liver metastasis of colon cancer organoids.This model is one of hepatic colonization and eventual outgrowth and not metastasis. Metastasis is optimally assessed using autochthonous models of cancer generation, with the concomitant intravasation, extravasation, and growth of cancer cells at the distant site. The authors should inject their various organoids in an orthotopic colonic transplantation assay, which permits the growth of tumors in the colon, and they can then identify metastasis in the liver that results from that primary cancer lesion (i.e., to better model physiologic metastasis from the colon to liver).

The data of orthotopic colonic transplantation data has been provided above (Author response images 1 and 2).

Author claim 2: DKK2 is required for the formation of lysozyme-positive cells in colon cancer.It would greatly strengthen the authors' claim if supraphysiologic or very high amounts of DKK2 enhance CRC organoid line engraftment (i.e., the specific experiment being pre-treatment with high levels of DKK2 and immediate transplantation to see a number of outgrowing clones). If DKK2 is causal for the engraftment of the tumors, increased DKK2 should enhance their capacity for engraftment.Paneth cells have physical properties permitting sorting and are readily identifiable on flow cytometry. The authors should demonstrate increased tumorigenicity and engraftment by sorting the Paneth-like cells-either by the markers they identified in the subsequent single cell or by scatter to establish whether the frequency of the Paneth-like cells in a culture of organoids is directly correlated with engraftment potential.Further characterization of the Paneth-like cells would help further the authors' argument for the unique function that they have in their tumor model. Specifically, do the cells with Paneth-like cells secrete Wnt3, EGF, Notch ligand, and DII4 as normal Paneth cells do? Immunofluorescence, sorting, or western blots would all be reasonable methods to assess protein levels in the sorted population.

This has been performed and provided above (Author response images 1 and 3)

Author claim 3: Lyzosome (LYZ)+ cancer cells exhibit Paneth cell properties in both mouse and human systems.For the claim to be general to human cancer, the author should demonstrate that loss of DKK2 impacts LYZ+ cancer cells in human organoids and affects their engraftment in immunodeficient mice compared to control. Another more correlative way to validate the LYZ+ expression in human colon cancer would be to stain for LYZ in metastatic vs. primary colon cancer, expecting metastatic lesions to be enriched for LYZ+ cells.The claims on the metabolic function of Paneth-like cells need more clarification. Do the cancer cells with Paneth features have a distinct metabolic profile compared to the other cell populations? The authors should address this through metabolic characterization of isolated LYZ+ cells with Seahorse or comparison of Dkk2 KO to WT organoids (i.e., +/-LYZ+ cancer cell population).

To address this question, we need to develop organoids with a Paneth cell reporter gene. We appreciate the reviewer’s comment, and this should be pursued in future studies.

Author claim 4: HNF4A mediates the formation of Lysozyme (Lyz)-positive colon cancer cells by DKK2.The authors implicate HNF4A and Sox9 as causal effectors of the Paneth-like cell phenotype and subsequent metastatic potential. There appears to be some discordance regarding the effect of DKK2 loss on HNF4A. In Figure 1E, the authors show that gene expression in metastatic colon cancer cells for HNF4A in DKK2 knockout vs AKP control is insignificant. However, in Figure 6I, there is a highly significant difference in the number of HNF4A positive cells, more than a 3-fold percentage difference, with a p-value of <0.0001. If there is the emergence of a rare but highly expressing HNF4A cell type that on aggregate bulk expression leads to no difference, but sorts differentially, why is it not identified in the single-cell data set? These data together are highly inconsistent with regards to the effect of DKK2 on HNF4A and require clarification.

Previous studies have demonstrated that HNF4A is regulated by proteasomal degradation mediated by pSrc. As a result, the mRNA level of HNF4A remains unchanged, while the protein level is significantly reduced in colon cancer cells. DKK2 KO leads to decreased Src phosphorylation, resulting in the recovery of HNF4A protein levels. This explains why HNF4A cannot be detected in scRNA-seq datasets, which measure mRNA. We have shown this in our previous report. In this manuscript, based on ChIP-seq data using an anti-HNF4A monoclonal antibody, as well as confocal microscopy and qPCR data for the Sox9 gene, we propose that HNF4A acts as a regulator of cancer cells exhibiting Paneth cell properties.